# Scalable Kernel Methods via Doubly Stochastic Gradients

**Bo Dai**[1], **Bo Xie**[1], **Niao He**[1], **Yingyu Liang**[2], **Anant Raj**[1], **Maria-Florina Balcan**[3], **Le Song**[1]

[1]Georgia Institute of Technology

{bodai, bxie33, nhe6, araj34}@gatech.edu, lsong@cc.gatech.edu

[2]Princeton University        [3]Carnegie Mellon University

yingyul@cs.princeton.edu        ninamf@cs.cmu.edu

## Abstract

The general perception is that kernel methods are not scalable, so neural nets become the choice for large-scale nonlinear learning problems. Have we tried hard enough for kernel methods? In this paper, we propose an approach that scales up kernel methods using a novel concept called "*doubly stochastic functional gradients*". Based on the fact that many kernel methods can be expressed as convex optimization problems, our approach solves the optimization problems by making *two unbiased* stochastic approximations to the functional gradient—one using random training points and another using random features associated with the kernel—and performing descent steps with this noisy functional gradient. Our algorithm is simple, need *no* commit to a preset number of random features, and allows the flexibility of the function class to grow as we see more incoming data in the streaming setting. We demonstrate that a function learned by this procedure after $t$ iterations converges to the optimal function in the reproducing kernel Hilbert space in rate $O(1/t)$, and achieves a generalization bound of $O(1/\sqrt{t})$. Our approach can readily scale kernel methods up to the regimes which are dominated by neural nets. We show competitive performances of our approach as compared to neural nets in datasets such as 2.3 million energy materials from MolecularSpace, 8 million handwritten digits from MNIST, and 1 million photos from ImageNet using convolution features.

## 1 Introduction

The general perception is that kernel methods are not scalable. When it comes to large-scale nonlinear learning problems, the methods of choice so far are neural nets although theoretical understanding remains incomplete. Are kernel methods really not scalable? Or is it simply because we have not tried hard enough, while neural nets have exploited sophisticated design of feature architectures, virtual example generation for dealing with invariance, stochastic gradient descent for efficient training, and GPUs for further speedup?

A bottleneck in scaling up kernel methods comes from the storage and computation cost of the dense kernel matrix, $K$. Storing the matrix requires $O(n^2)$ space, and computing it takes $O(n^2 d)$ operations, where $n$ is the number of data points and $d$ is the dimension. There have been many great attempts to scale up kernel methods, including efforts in perspectives of numerical linear algebra, functional analysis, and numerical optimization.

A common numerical linear algebra approach is to approximate the kernel matrix using low-rank factorizations, $K \approx A^\top A$, with $A \in \mathbb{R}^{r \times n}$ and rank $r \leqslant n$. This low-rank approximation allows subsequent kernel algorithms to directly operate on $A$, but computing the approximation requires $O(nr^2 + nrd)$ operations. Many work followed this strategy, including Greedy basis selection techniques [1], Nyström approximation [2] and incomplete Cholesky decomposition [3]. In practice, one observes that kernel methods with approximated kernel matrices often result in a few percentage of losses in performance. In fact, without further assumption on the regularity of the

kernel matrix, the generalization ability after using low-rank approximation is typically of order $O(1/\sqrt{r} + 1/\sqrt{n})$ [4, 5], which implies that the rank needs to be nearly linear in the number of data points! Thus, in order for kernel methods to achieve the best generalization ability, low-rank approximation based approaches immediately become impractical for big datasets because of their $O(n^3 + n^2 d)$ preprocessing time and $O(n^2)$ storage.

Random feature approximation is another popular approach for scaling up kernel methods [6, 7]. The method directly approximates the kernel function instead of the kernel matrix using explicit feature maps. The advantage of this approach is that the random feature matrix for $n$ data points can be computed in time $O(nrd)$ using $O(nr)$ storage, where $r$ is the number of random features. Subsequent algorithms then only need to operate on an $O(nr)$ matrix. Similar to low-rank kernel matrix approximation approach, the generalization ability of this approach is of the order $O(1/\sqrt{r} + 1/\sqrt{n})$ [8, 9], which implies that the number of random features also needs to be $O(n)$. Another common drawback of these two approaches is that adapting the solution from a small $r$ to a large $r'$ is not easy if one wants to increase the rank of the approximated kernel matrix or the number of random features for better generalization ability. Special procedures need to be designed to reuse the solution obtained from a small $r$, which is not straightforward.

Another approach that addresses the scalability issue rises from the optimization perspective. One general strategy is to solve the dual forms of kernel methods using the block-coordinate descent (*e.g.*, [10, 11, 12]). Each iteration of this algorithm only incurs $O(nrd)$ computation and $O(nr)$ storage, where $r$ is the block size. A second strategy is to perform functional gradient descent based on a batch of data points at each epoch (*e.g.*, [13, 14]). Thus, the computation and storage in each iteration required are also $O(nrd)$ and $O(nr)$, respectively, where $r$ is the batch size. These approaches can straightforwardly adapt to a different $r$ without restarting the optimization procedure and exhibit no generalization loss since they do not approximate the kernel matrix or function. However, a serious drawback of these approaches is that, without further approximation, all support vectors need to be stored for testing, which can be as big as the entire training set! (*e.g.*, kernel ridge regression and non-separable nonlinear classification problems.)

In summary, there exists a delicate trade-off between computation, storage and statistics when scaling up kernel methods. Inspired by various previous efforts, we propose a simple yet general strategy that scales up many kernel methods using a novel concept called "*doubly stochastic functional gradients*". Our method relies on the fact that most kernel methods can be expressed as convex optimization problems over functions in the reproducing kernel Hilbert spaces (RKHS) and solved via functional gradient descent. Our algorithm proceeds by making *two unbiased* stochastic approximations to the functional gradient, one using random training points and another using random functions associated with the kernel, and then descending using this noisy functional gradient. The key intuitions behind our algorithm originate from **(i)** the property of stochastic gradient descent algorithm that as long as the stochastic gradient is unbiased, the convergence of the algorithm is guaranteed [15]; and **(ii)** the property of pseudo-random number generators that the random samples can in fact be completely determined by an initial value (a seed). We exploit these properties and enable kernel methods to achieve better balances between computation, storage, and statistics. Our method interestingly integrates kernel methods, functional analysis, stochastic optimization, and algorithmic tricks, and it possesses a number of desiderata:

**Generality and simplicity.** Our approach applies to many kernel methods such as kernel version of ridge regression, support vector machines, logistic regression and two-sample test as well as many different types of kernels such as shift-invariant, polynomial, and general inner product kernels. The algorithm can be summarized in just a few lines of code (Algorithm 1 and 2). For a different problem and kernel, we just need to replace the loss function and the random feature generator.

**Flexibility.** While previous approaches based on random features typically require a prefix number of features, our approach allows the number of random features, and hence the flexibility of the function class to grow with the number of data points. Therefore, unlike previous random feature approach, our approach applies to the data streaming setting and achieves full potentials of nonparametric methods.

**Efficient computation.** The key computation of our method comes from evaluating the doubly stochastic functional gradient, which involves the generation of the random features given specific seeds and also the evaluation of these features on a small batch of data points. At iteration $t$, the computational complexity is $O(td)$.

**Small memory.** While most approaches require saving all the support vectors, the algorithm allows us to avoid keeping the support vectors since it only requires a small program to regenerate the random features and sample historical features according to some specific random seeds. At iteration $t$, the memory needed is $O(t)$, independent of the dimension of the data.

**Theoretical guarantees.** We provide novel and nontrivial analysis involving Hilbert space martingales and a newly proved recurrence relation, and demonstrate that the estimator produced by our algorithm, which might be outside of the RKHS, converges to the optimal RKHS function. More specifically, both in expectation and with high probability, our algorithm estimates the optimal function in the RKHS in the rate of $O(1/t)$ and achieves a generalization bound of $O(1/\sqrt{t})$, which are indeed optimal [15]. The variance of the random features introduced in our second approximation to the functional gradient, only contributes additively to the constant in the convergence rate. These results are the first of the kind in literature, which could be of independent interest.

**Strong empirical performance.** Our algorithm can readily scale kernel methods up to the regimes which are previously dominated by neural nets. We show that our method compares favorably to other scalable kernel methods in medium scale datasets, and to neural nets in big datasets with millions of data.

In the remainder, we will first introduce preliminaries on kernel methods and functional gradients. We will then describe our algorithm and provide both theoretical and empirical supports.

## 2 Duality between Kernels and Random Processes

Kernel methods owe their name to the use of kernel functions, $k(x, x') : \mathcal{X} \times \mathcal{X} \mapsto \mathbb{R}$, which are symmetric positive definite (PD), meaning that for all $n > 1$, and $x_1, \ldots, x_n \in \mathcal{X}$, and $c_1, \ldots, c_n \in \mathbb{R}$, we have $\sum_{i,j=1}^{n} c_i c_j k(x_i, x_j) \geqslant 0$. There is an intriguing duality between kernels and stochastic processes which will play a crucial role in our algorithm design later. More specifically,

**Theorem 1 (*e.g.*, Devinatz [16]; Hein & Bousquet [17])** *If $k(x, x')$ is a PD kernel, then there exists a set $\Omega$, a measure $\mathbb{P}$ on $\Omega$, and random function $\phi_\omega(x) : \mathcal{X} \mapsto \mathbb{R}$ from $L_2(\Omega, \mathbb{P})$, such that $k(x, x') = \int_\Omega \phi_\omega(x) \phi_\omega(x') \, d\mathbb{P}(\omega)$.*

Essentially, the above integral representation relates the kernel function to a random process $\omega$ with measure $\mathbb{P}(\omega)$. Note that the integral representation may not be unique. For instance, the random process can be a Gaussian process on $\mathcal{X}$ with the sample function $\phi_\omega(x)$, and $k(x, x')$ is simply the covariance function between two point $x$ and $x'$. If the kernel is also continuous and shift invariant, *i.e.*, $k(x, x') = k(x - x')$ for $x \in \mathbb{R}^d$, then the integral representation specializes into a form characterized by inverse Fourier transformation (*e.g.*, [18, Theorem 6.6]),

**Theorem 2 (Bochner)** *A continuous, real-valued, symmetric and shift-invariant function $k(x - x')$ on $\mathbb{R}^d$ is a PD kernel if and only if there is a finite non-negative measure $\mathbb{P}(\omega)$ on $\mathbb{R}^d$, such that $k(x - x') = \int_{\mathbb{R}^d} e^{i\omega^\top(x-x')} \, d\mathbb{P}(\omega) = \int_{\mathbb{R}^d \times [0,2\pi]} 2 \cos(\omega^\top x + b) \cos(\omega^\top x' + b) \, d\left(\mathbb{P}(\omega) \times \mathbb{P}(b)\right)$, where $\mathbb{P}(b)$ is a uniform distribution on $[0, 2\pi]$, and $\phi_\omega(x) = \sqrt{2} \cos(\omega^\top x + b)$.*

For Gaussian RBF kernel, $k(x - x') = \exp(-\|x - x'\|^2 / 2\sigma^2)$, this yields a Gaussian distribution $\mathbb{P}(\omega)$ with density proportional to $\exp(-\sigma^2 \|\omega\|^2 / 2)$; for the Laplace kernel, this yields a Cauchy distribution; and for the Martern kernel, this yields the convolutions of the unit ball [19]. Similar representations where the explicit form of $\phi_\omega(x)$ and $\mathbb{P}(\omega)$ are known can also be derived for rotation invariant kernel, $k(x, x') = k(\langle x, x' \rangle)$, using Fourier transformation on sphere [19]. For polynomial kernels, $k(x, x') = (\langle x, x' \rangle + c)^p$, a random tensor sketching approach can also be used [20].

Instead of finding the random processes $\mathbb{P}(\omega)$ and functions $\phi_\omega(x)$ given kernels, one can go the reverse direction and construct kernels from random processes and functions (*e.g.*, Wendland [18]).

**Theorem 3** *If $k(x, x') = \int_\Omega \phi_\omega(x) \phi_\omega(x') \, d\mathbb{P}(\omega)$ for a nonnegative measure $\mathbb{P}(\omega)$ on $\Omega$ and $\phi_\omega(x) : \mathcal{X} \mapsto \mathbb{R}$ from $L_2(\Omega, \mathbb{P})$, then $k(x, x')$ is a PD kernel.*

For instance, $\phi_\omega(x) := \cos(\omega^\top \psi_\theta(x) + b)$, where $\psi_\theta(x)$ can be a random convolution of the input $x$ parametrized by $\theta$. Another important concept is the reproducing kernel Hilbert space (RKHS). An RKHS $\mathcal{H}$ on $\mathcal{X}$ is a Hilbert space of functions from $\mathcal{X}$ to $\mathbb{R}$. $\mathcal{H}$ is an RKHS if and only if there exists a $k(x, x') : \mathcal{X} \times \mathcal{X} \mapsto \mathbb{R}$ such that $\forall x \in \mathcal{X}, k(x, \cdot) \in \mathcal{H}$, and $\forall f \in \mathcal{H}, \langle f(\cdot), k(x, \cdot) \rangle_\mathcal{H} = f(x)$. If such a $k(x, x')$ exists, it is unique and it is a PD kernel. A function $f \in \mathcal{H}$ if and only if $\|f\|_\mathcal{H}^2 := \langle f, f \rangle_\mathcal{H} < \infty$, and its $L_2$ norm is dominated by RKHS norm, $\|f\|_{L_2} \leqslant \|f\|_\mathcal{H}$.

## 3 Doubly Stochastic Functional Gradients

Many kernel methods can be written as convex optimization problems over functions in the RKHS and solved using the functional gradient methods [13, 14]. Inspired by these previous work, we will introduce a novel concept called "*doubly stochastic functional gradients*" to address the scalability issue. Let $l(u, y)$ be a scalar loss function convex of $u \in \mathbb{R}$. Let the subgradient of $l(u, y)$ with respect to $u$ be $l'(u, y)$. Given a PD kernel $k(x, x')$ and the associated RKHS $\mathcal{H}$, many kernel methods try to find a function $f_* \in \mathcal{H}$ which solves the optimization problem

$$\underset{f \in \mathcal{H}}{\operatorname{argmin}} \quad R(f) := \mathbb{E}_{(x,y)}[l(f(x), y)] + \frac{\nu}{2} \|f\|_{\mathcal{H}}^2 \quad \Longleftrightarrow \quad \underset{\|f\|_{\mathcal{H}} \leqslant B(\nu)}{\operatorname{argmin}} \quad \mathbb{E}_{(x,y)}[l(f(x), y)] \quad (1)$$

where $\nu > 0$ is a regularization parameter, $B(\nu)$ is a non-increasing function of $\nu$, and the data $(x, y)$ follow a distribution $\mathbb{P}(x, y)$. The functional gradient $\nabla R(f)$ is defined as the linear term in the change of the objective after we perturb $f$ by $\epsilon$ in the direction of $g$, *i.e.*,

$$R(f + \epsilon g) = R(f) + \epsilon \langle \nabla R(f), g \rangle_{\mathcal{H}} + O(\epsilon^2). \quad (2)$$

For instance, applying the above definition, we have $\nabla f(x) = \nabla \langle f, k(x, \cdot) \rangle_{\mathcal{H}} = k(x, \cdot)$, and $\nabla \|f\|_{\mathcal{H}}^2 = \nabla \langle f, f \rangle_{\mathcal{H}} = 2f$.

**Stochastic functional gradient.** Given a data point $(x, y) \sim \mathbb{P}(x, y)$ and $f \in \mathcal{H}$, the stochastic functional gradient of $\mathbb{E}_{(x,y)}[l(f(x), y)]$ with respect to $f \in \mathcal{H}$ is

$$\xi(\cdot) := l'(f(x), y)k(x, \cdot), \quad (3)$$

which is essentially a single data point approximation to the true functional gradient. Furthermore, for any $g \in \mathcal{H}$, we have $\langle \xi(\cdot), g \rangle_{\mathcal{H}} = l'(f(x), y)g(x)$. Inspired by the duality between kernel functions and random processes, we can make an additional approximation to the stochastic functional gradient using a random function $\phi_\omega(x)$ sampled according to $\mathbb{P}(\omega)$. More specifically,

**Doubly stochastic functional gradient.** Let $\omega \sim \mathbb{P}(\omega)$, then the doubly stochastic gradient of $\mathbb{E}_{(x,y)}[l(f(x), y)]$ with respect to $f \in \mathcal{H}$ is

$$\zeta(\cdot) := l'(f(x), y)\phi_\omega(x)\phi_\omega(\cdot). \quad (4)$$

Note that the stochastic functional gradient $\xi(\cdot)$ is in RKHS $\mathcal{H}$ but $\zeta(\cdot)$ may be outside $\mathcal{H}$, since $\phi_\omega(\cdot)$ may be outside the RKHS. For instance, for the Gaussian RBF kernel, the random function $\phi_\omega(x) = \sqrt{2} \cos(\omega^\top x + b)$ is outside the RKHS associated with the kernel function.

However, these functional gradients are related by $\xi(\cdot) = \mathbb{E}_\omega[\zeta(\cdot)]$, which lead to unbiased estimators of the original functional gradient, *i.e.*,

$$\nabla R(f) = \mathbb{E}_{(x,y)}[\xi(\cdot)] + \nu f(\cdot), \quad \text{and} \quad \nabla R(f) = \mathbb{E}_{(x,y)}\mathbb{E}_\omega[\zeta(\cdot)] + \nu f(\cdot). \quad (5)$$

We emphasize that the source of randomness associated with the random function is not present in the data, but artificially introduced by us. This is crucial for the development of our scalable algorithm in the next section. Meanwhile, it also creates additional challenges in the analysis of the algorithm which we will deal with carefully.

## 4 Doubly Stochastic Kernel Machines

---
**Algorithm 1:** $\{\alpha_i\}_{i=1}^t = \textbf{Train}(\mathbb{P}(x, y))$

---
**Require:** $\mathbb{P}(\omega)$, $\phi_\omega(x)$, $l(f(x), y)$, $\nu$.
  1: **for** $i = 1, \ldots, t$ **do**
  2:     Sample $(x_i, y_i) \sim \mathbb{P}(x, y)$.
  3:     Sample $\omega_i \sim \mathbb{P}(\omega)$ with seed $i$.
  4:     $f(x_i) = \textbf{Predict}(x_i, \{\alpha_j\}_{j=1}^{i-1})$.
  5:     $\alpha_i = -\gamma_i l'(f(x_i), y_i)\phi_{\omega_i}(x_i)$.
  6:     $\alpha_j = (1 - \gamma_i \nu)\alpha_j$ for $j = 1, \ldots, i - 1$.
  7: **end for**

---
**Algorithm 2:** $f(x) = \textbf{Predict}(x, \{\alpha_i\}_{i=1}^t)$

---
**Require:** $\mathbb{P}(\omega)$, $\phi_\omega(x)$.
  1: Set $f(x) = 0$.
  2: **for** $i = 1, \ldots, t$ **do**
  3:     Sample $\omega_i \sim \mathbb{P}(\omega)$ with seed $i$.
  4:     $f(x) = f(x) + \alpha_i \phi_{\omega_i}(x)$.
  5: **end for**

---

The first key intuition behind our algorithm originates from the property of stochastic gradient descent algorithm that as long as the stochastic gradient is bounded and unbiased, the convergence of the algorithm is guaranteed [15]. In our algorithm, we will exploit this property and introduce *two* sources of randomness, one from data and another artificial, to scale up kernel methods.

The second key intuition behind our algorithm is that the random functions used in the doubly stochastic functional gradients will be sampled according to *pseudo-random number generators*, where the sequences of apparently random samples can in fact be completely determined by an initial value (a seed). Although these random samples are not the "true" random sample in the purest sense of the word, they suffice for our task in practice.

To be more specific, our algorithm proceeds by making two stochastic approximation to the functional gradient in each iteration, and then descending using this noisy functional gradient. The overall algorithms for training and prediction are summarized in Algorithm 1 and 2. The training algorithm essentially just performs samplings of random functions and evaluations of doubly stochastic gradients and maintains a collection of real numbers $\{\alpha_i\}$, which is computationally efficient and memory friendly. A crucial step in the algorithm is to sample the random functions with "*seed i*". The seeds have to be aligned between training and prediction, and with the corresponding $\alpha_i$ obtained from each iteration. The learning rate $\gamma_t$ in the algorithm needs to be chosen as $O(1/t)$, as shown by our later analysis to achieve the best rate of convergence. For now, we assume that we have access to the data generating distribution $\mathbb{P}(x, y)$. This can be modified to sample uniformly randomly from a fixed dataset, without affecting the algorithm and the later convergence analysis. Let the sampled data and random function parameters be $\mathcal{D}^t := \{(x_i, y_i)\}_{i=1}^t$ and $\boldsymbol{\omega}^t := \{\omega_i\}_{i=1}^t$, respectively after $t$ iteration. The function obtained by Algorithm 1 is a simple additive form of the doubly stochastic functional gradients

$$f_{t+1}(\cdot) = f_t(\cdot) - \gamma_t(\zeta_t(\cdot) + \nu f_t(\cdot)) = \sum_{i=1}^t a_t^i \zeta_i(\cdot), \quad \forall t > 1, \quad \text{and} \quad f_1(\cdot) = 0, \qquad (6)$$

where $a_t^i = -\gamma_i \prod_{j=i+1}^t (1 - \gamma_j \nu)$ are deterministic values depending on the step sizes $\gamma_j (i \leqslant j \leqslant t)$ and regularization parameter $\nu$. This simple form makes it easy for us to analyze its convergence.

We note that our algorithm can also take a mini-batch of points and random functions at each step, and estimate an empirical covariance for preconditioning to achieve potentially better performance.

## 5 Theoretical Guarantees

In this section, we will show that, both in expectation and with high probability, our algorithm can estimate the optimal function in the RKHS with rate $O(1/t)$ and achieve a generalization bound of $O(1/\sqrt{t})$. The analysis for our algorithm has a new twist compared to previous analysis of stochastic gradient descent algorithms, since the random function approximation results in an estimator which is outside the RKHS. Besides the analysis for stochastic functional gradient descent, we need to use martingales and the corresponding concentration inequalities to prove that the sequence of estimators, $f_{t+1}$, outside the RKHS converge to the optimal function, $f_*$, in the RKHS. We make the following standard assumptions ahead for later references:

A. There exists an optimal solution, denoted as $f_*$, to the problem of our interest (1).
B. Loss function $\ell(u, y) : \mathbb{R} \times \mathbb{R} \to \mathbb{R}$ and its first-order derivative is $L$-Lipschitz continous in terms of the first argument.
C. For any data $\{(x_i, y_i)\}_{i=1}^t$ and any trajectory $\{f_i(\cdot)\}_{i=1}^t$, there exists $M > 0$, such that $|\ell'(f_i(x_i), y_i)| \leqslant M$. Note in our situation $M$ exists and $M < \infty$ since we assume bounded domain and the functions $f_t$ we generate are always bounded as well.
D. There exists $\kappa > 0$ and $\phi > 0$, such that $k(x, x') \leqslant \kappa$, $|\phi_\omega(x)\phi_\omega(x')| \leqslant \phi, \forall x, x' \in \mathcal{X}, \omega \in \Omega$. For example, when $k(\cdot, \cdot)$ is the Gaussian RBF kernel, we have $\kappa = 1, \phi = 2$.

We now present our main theorems as below. Due to the space restrictions, we will only provide a short sketch of proofs here. The full proofs for the these theorems are given in the appendix.

**Theorem 4 (Convergence in expectation)** *When $\gamma_t = \frac{\theta}{t}$ with $\theta > 0$ such that $\theta\nu \in (1, 2) \cup \mathbb{Z}_+$,*

$$\mathbb{E}_{\mathcal{D}^t, \boldsymbol{\omega}^t}\left[|f_{t+1}(x) - f_*(x)|^2\right] \leqslant \frac{2C^2 + 2\kappa Q_1^2}{t}, \quad \text{for any } x \in \mathcal{X}$$

*where $Q_1 = \max\left\{\|f_*\|_{\mathcal{H}}, (Q_0 + \sqrt{Q_0^2 + (2\theta\nu - 1)(1 + \theta\nu)^2\theta^2\kappa M^2})/(2\nu\theta - 1)\right\}$, with $Q_0 = 2\sqrt{2}\kappa^{1/2}(\kappa + \phi)LM\theta^2$, and $C^2 = 4(\kappa + \phi)^2 M^2 \theta^2$.*

**Theorem 5 (Convergence with high probability)** *When $\gamma_t = \frac{\theta}{t}$ with $\theta > 0$ such that $\theta\nu \in \mathbb{Z}_+$, for any $x \in \mathcal{X}$, we have with probability at least $1 - 3\delta$ over $(\mathcal{D}^t, \boldsymbol{\omega}^t)$,*

$$|f_{t+1}(x) - f_*(x)|^2 \leqslant \frac{C^2 \ln(2/\delta)}{t} + \frac{2\kappa Q_2^2 \ln(2t/\delta) \ln^2(t)}{t},$$

*where $C$ is as above and $Q_2 = \max\left\{\|f_*\|_{\mathcal{H}}, Q_0 + \sqrt{Q_0^2 + \kappa M^2(1+\theta\nu)^2(\theta^2 + 16\theta/\nu)}\right\}$, with*
$Q_0 = 4\sqrt{2}\kappa^{1/2}M\theta(8 + (\kappa + \phi)\theta L).$

**Proof sketch:** We focus on the convergence in expectation; the high probability bound can be established in a similar fashion. The main technical difficulty is that $f_{t+1}$ may not be in the RKHS $\mathcal{H}$. The key of the proof is then to construct an intermediate function $h_{t+1}$, such that the difference between $f_{t+1}$ and $h_{t+1}$ and the difference between $h_{t+1}$ and $f_*$ can be bounded. More specifically,

$$h_{t+1}(\cdot) = h_t(\cdot) - \gamma_t(\xi_t(\cdot) + \nu h_t(\cdot)) = \sum_{i=1}^{t} a_t^i \xi_i(\cdot), \quad \forall t > 1, \quad \text{and} \quad h_1(\cdot) = 0, \quad (7)$$

where $\xi_t(\cdot) = \mathbb{E}_{\omega_t}[\zeta_t(\cdot)]$. Then for any $x$, the error can be decomposed as two terms

$$|f_{t+1}(x) - f_*(x)|^2 \leqslant 2 \underbrace{|f_{t+1}(x) - h_{t+1}(x)|^2}_{\text{error due to random functions}} + 2\kappa \underbrace{\|h_{t+1} - f_*\|_{\mathcal{H}}^2}_{\text{error due to random data}}$$

For the error term due to random functions, $h_{t+1}$ is constructed such that $f_{t+1} - h_{t+1}$ is a martingale, and the stepsizes are chosen such that $|a_t^i| \leqslant \frac{\theta}{t}$, which allows us to bound the martingale. In other words, the choices of the stepsizes keep $f_{t+1}$ close to the RKHS. For the error term due to random data, since $h_{t+1} \in \mathcal{H}$, we can now apply the standard arguments for stochastic approximation in the RKHS. Due to the additional randomness, the recursion is slightly more complicated, $e_{t+1} \leqslant \left(1 - \frac{2\nu\theta}{t}\right)e_t + \frac{\beta_1}{t}\sqrt{\frac{e_t}{t}} + \frac{\beta_2}{t^2}$, where $e_{t+1} = \mathbb{E}_{\mathcal{D}^t, \omega^t}[\|h_{t+1} - f_*\|_{\mathcal{H}}^2]$, and $\beta_1$ and $\beta_2$ depends on the related parameters. Solving this recursion then leads to a bound for the second error term. ∎

**Theorem 6 (Generalization bound)** *Let the true risk be $R_{true}(f) = \mathbb{E}_{(x,y)}[l(f(x),y)]$. Then with probability at least $1 - 3\delta$ over $(\mathcal{D}^t, \omega^t)$, and $C$ and $Q_2$ defined as previously*

$$R_{true}(f_{t+1}) - R_{true}(f_*) \leqslant \frac{(C\sqrt{\ln(8\sqrt{e}t/\delta)} + \sqrt{2\kappa}Q_2\sqrt{\ln(2t/\delta)}\ln(t))L}{\sqrt{t}}.$$

**Proof** By the Lipschitz continuity of $l(\cdot, y)$ and Jensen's Inequality, we have

$$R_{true}(f_{t+1}) - R_{true}(f_*) \leqslant L\mathbb{E}_x|f_{t+1}(x) - f_*(x)| \leqslant L\sqrt{\mathbb{E}_x|f_{t+1}(x) - f_*(x)|^2} = L\|f_{t+1} - f_*\|_2.$$

Again, $\|f_{t+1} - f_*\|_2$ can be decomposed as two terms $O\left(\|f_{t+1} - h_{t+1}\|_2^2\right)$ and $O(\|h_{t+1} - f_*\|_{\mathcal{H}}^2)$, which can be bounded similarly as in Theorem 5 (see Corollary 12 in the appendix). ∎

**Remarks.** The overall rate of convergence in expectation, which is $O(1/t)$, is indeed optimal. Classical complexity theory (see, e.g. reference in [15]) shows that to obtain $\epsilon$-accuracy solution, the number of iterations needed for the stochastic approximation is $\Omega(1/\epsilon)$ for strongly convex case and $\Omega(1/\epsilon^2)$ for general convex case. Different from the classical setting of stochastic approximation, our case imposes not one but two sources of randomness/stochasticity in the gradient, which intuitively speaking, might require higher order number of iterations for general convex case. However, our method is still able to achieve the same rate as in the classical setting. The rate of the generalization bound is also nearly optimal up to log factors. However, these bounds may be further refined with more sophisticated techniques and analysis. For example, mini-batch and preconditioning can be used to reduce the constant factors in the bound significantly, the analysis of which is left for future study. Theorem 4 also reveals bounds in $L_\infty$ and $L_2$ sense as in Section A.2 in the appendix. The choices of stepsizes $\gamma_t$ and the tuning parameters given in these bounds are only for sufficient conditions and simple analysis; other choices can also lead to bounds in the same order.

## 6  Computation, Storage and Statistics Trade-off

To investigate computation, storage and statistics trade-off, we will fix the desired $L_2$ error in the function estimation to $\epsilon$, *i.e.*, $\|f - f_*\|_2^2 \leqslant \epsilon$, and work out the dependency of other quantities on $\epsilon$. These other quantities include the preprocessing time, the number of samples and random features (or rank), the number of iterations of each algorithm, and the computational cost and storage requirement for learning and prediction. We assume that the number of samples, $n$, needed to achieve the prescribed error $\epsilon$ is of the order $O(1/\epsilon)$, the same for all methods. Furthermore, we make no other regularity assumption about margin properties or the kernel matrix such as fast spectrum decay. Thus the required number of random feature (or ranks) $r$ will be of the order $O(n) = O(1/\epsilon)$ [4, 5, 8, 9].

We will pick a few representative algorithms for comparison, namely, *(i)* NORMA [13]: kernel methods trained with stochastic functional gradients; *(ii)* k-SDCA [12]: kernelized version of stochastic dual coordinate ascend; *(iii)* r-SDCA: first approximate the kernel function with random features, and then run stochastic dual coordinate ascend; *(iv)* n-SDCA: first approximate the kernel matrix using Nyström's method, and then run stochastic dual coordinate ascend; similarly we will combine Pegasos algorithm [21] with random features and Nyström's method, and obtain *(v)* r-Pegasos, and *(vi)* n-Pegasos. The comparisons are summarized below.

From the table, one can see that our method, r-SDCA and r-Pegasos achieve the best dependency on the dimension $d$ of the data. However, often one is interested in increasing the number of random features as more data points are observed to obtain a better generalization ability. Then special procedures need to be designed for updating the r-SDCA and r-Pegasos solution, which we are not clear how to implement easily and efficiently.

| Algorithms | Preprocessing Computation | Total Computation Cost | | Total Storage Cost | |
|---|---|---|---|---|---|
| | | Training | Prediction | Training | Prediction |
| Doubly SGD | $O(1)$ | $O(d/\epsilon^2)$ | $O(d/\epsilon)$ | $O(1/\epsilon)$ | $O(1/\epsilon)$ |
| NORMA/k-SDCA | $O(1)$ | $O(d/\epsilon^2)$ | $O(d/\epsilon)$ | $O(d/\epsilon)$ | $O(d/\epsilon)$ |
| r-Pegasos/r-SDCA | $O(1)$ | $O(d/\epsilon^2)$ | $O(d/\epsilon)$ | $O(1/\epsilon)$ | $O(1/\epsilon)$ |
| n-Pegasos/n-SDCA | $O(1/\epsilon^3)$ | $O(d/\epsilon^2)$ | $O(d/\epsilon)$ | $O(1/\epsilon)$ | $O(1/\epsilon)$ |

# 7 Experiments

We show that our method compares favorably to other kernel methods in medium scale datasets and neural nets in large scale datasets. We examined both regression and classification problems with smooth and almost smooth loss functions. Below is a summary of the datasets used[1], and more detailed description of these datasets and experimental settings can be found in the appendix.

| | Name | Model | # of samples | Input dim | Output range | Virtual |
|---|---|---|---|---|---|---|
| (1) | Adult | K-SVM | 32K | 123 | $\{-1, 1\}$ | no |
| (2) | MNIST 8M 8 vs. 6 [25] | K-SVM | 1.6M | 784 | $\{-1, 1\}$ | yes |
| (3) | Forest | K-SVM | 0.5M | 54 | $\{-1, 1\}$ | no |
| (4) | MNIST 8M [25] | K-logistic | 8M | 1568 | $\{0, \ldots, 9\}$ | yes |
| (5) | CIFAR 10 [26] | K-logistic | 60K | 2304 | $\{0, \ldots, 9\}$ | yes |
| (6) | ImageNet [27] | K-logistic | 1.3M | 9216 | $\{0, \ldots, 999\}$ | yes |
| (7) | QuantumMachine [28] | K-ridge | 6K | 276 | $[-800, -2000]$ | yes |
| (8) | MolecularSpace [28] | K-ridge | 2.3M | 2850 | $[0, 13]$ | no |

**Experiment settings.** For datasets (1) – (3), we compare the algorithms discussed in Section 6. For algorithms based on low rank kernel matrix approximation and random features, *i.e.*, pegasos and SDCA, we set the rank and number of random features to be $2^8$. We use same batch size for both our algorithm and the competitors. We stop algorithms when they pass through the entire dataset once. This stopping criterion (SC1) is designed for justifying our conjecture that the bottleneck of the performances of the vanilla methods with explicit feature comes from the accuracy of kernel approximation. To this end, we investigate the performances of these algorithms under different levels of random feature approximations but within the same number of training samples. To further investigate the computational efficiency of the proposed algorithm, we also conduct experiments where we stop all algorithms within the same time budget (SC2). Due to space limitation, the comparison on regression synthetic dataset under SC1 and on (1) – (3) under SC2 are illustrated in Appendix B.2. We do not count the preprocessing time of Nyström's method for n-Pegasos and n-SDCA. The algorithms are executed on the machine with AMD 16 2.4GHz Opteron CPUs and 200G memory. Note that this allows NORMA and k-SDCA to save all the data in the memory.

We report our numerical results in Figure 1(1)-(8) with explanations stated as below . For full details of our experimental setups, please refer to section B.1 in Appendix.

**Adult.** The result is illustrated in Figure 1(1). NORMA and k-SDCA achieve the best error rate, $15\%$, while our algorithm achieves a comparable rate, $15.3\%$.

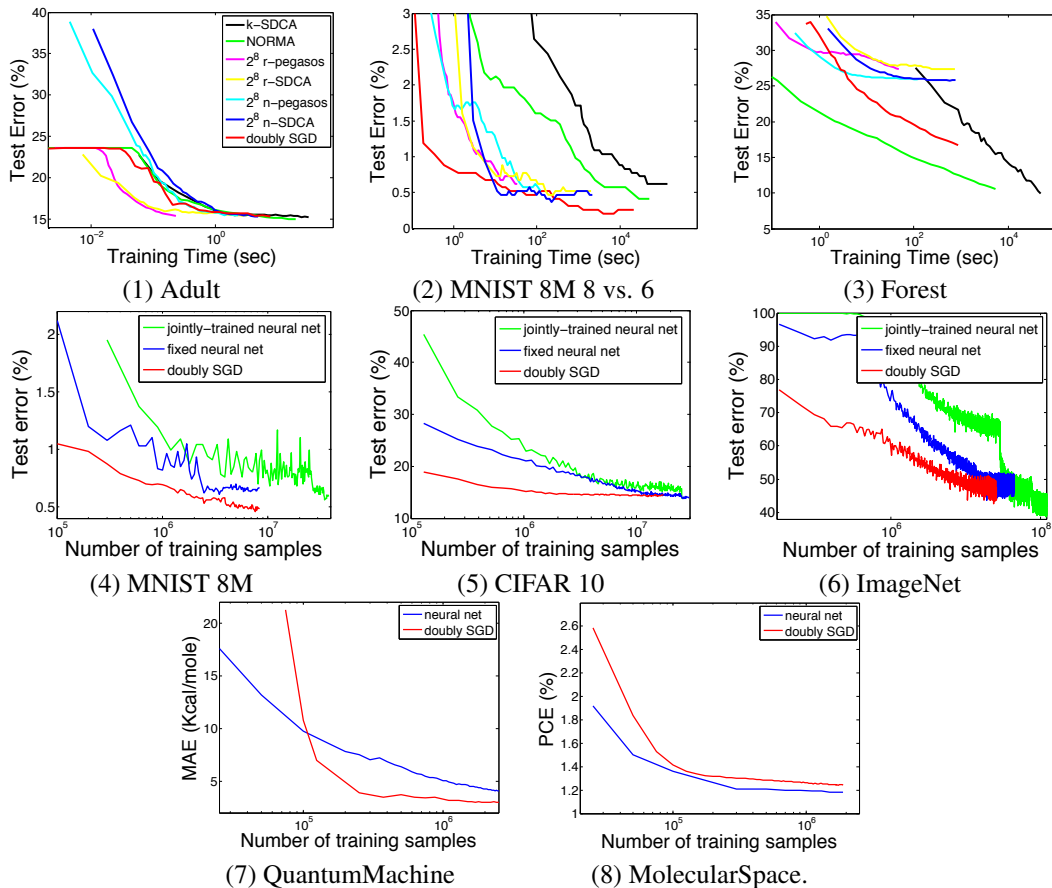

Figure 1: Experimental results for dataset (1) – (8).

**MNIST 8M 8 vs. 6.** The result is shown in Figure 1(2). Our algorithm achieves the best test error 0.26%. Comparing to the methods with full kernel, the methods using random/Nyström features achieve better test errors probably because of the underlying low-rank structure of the dataset.

**Forest.** The result is shown in Figure 1(3). Our algorithm achieves test error about 15%, much better than the n/r-pegasos and n/r-SDCA. Our method is preferable for this scenario, *i.e.*, huge datasets with sophisticated decision boundary considering the trade-off between cost and accuracy.

**MNIST 8M.** The result is shown in Figure 1(4). Better than the 0.6% error provided by fixed and jointly-trained neural nets, our method reaches an error of 0.5% very quickly.

**CIFAR 10** The result is shown in Figure 1(5). We compare our algorithm to a neural net with two convolution layers (after contrast normalization and max-pooling layers) and two local layers achieving 11% test error. The specification is at https://code.google.com/p/cuda-convnet/. Our method achieves comparable performance but much faster.

**ImageNet** The result is shown in Figure 1(6). Our method achieves test error 44.5% by further max-voting of 10 transformations of the test set while the jointly-trained neural net arrives at 42% (without variations in color and illumination), and the fixed neural net only achieves 46% with max-voting.

**QuantumMachine/MolecularSpace** The results are shown in Figure 1(7) &(8). On dataset (7), our method achieves Mean Absolute Error of 2.97 kcal/mole, outperforming neural nets, 3.51 kcal/mole, which is close to the 1 kcal/mole required for chemical accuracy. Moreover, the comparison on dataset (8) is the first in the literature, and our method is still comparable with neural net.

## Acknowledgement

M.B. is suppoerted in part by NSF CCF-0953192, CCF-1451177, CCF-1101283, and CCF-1422910, ONR N00014-09-1-0751, and AFOSR FA9550-09-1-0538. L.S. is supported in part by NSF IIS-1116886, NSF/NIH BIGDATA 1R01GM108341, NSF CAREER IIS-1350983, and a Raytheon Faculty Fellowship.

## Footnotes

[1] A "yes" for the last column means that virtual examples are generated from for training. K-ridge stands for kernel ridge regression; K-SVM stands for kernel SVM; K-logistic stands for kernel logistic regression.

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
