[Supplementary Material]

# Appendix

## A  Proof Details

### A.1  Convergence Rate

We first provide specific bounds and detailed proofs for the two error terms appeared in Theorem 4 and Theorem 5.

#### A.1.1  Error due to random features

**Lemma 7** *We have*

(i) *For any $x \in \mathcal{X}$, $\mathbb{E}_{\mathcal{D}^t, \boldsymbol{\omega}^t}[|f_{t+1}(x) - h_{t+1}(x)|^2] \leqslant B_{1,t+1}^2 := 4M^2(\kappa + \phi)^2 \sum_{i=1}^t |a_t^i|^2$.*

(ii) *For any $x \in \mathcal{X}$, with probability at least $1 - \delta$ over $(\mathcal{D}^t, \boldsymbol{\omega}^t)$,*

$$|f_{t+1}(x) - h_{t+1}(x)|^2 \leqslant B_{2,t+1}^2 := 2M^2(\kappa + \phi)^2 \ln\left(\frac{2}{\delta}\right) \sum_{i=1}^t |a_t^i|^2$$

**Proof**  Let $V_i(x) = V_i(x; \mathcal{D}^i, \boldsymbol{\omega}^i) := a_t^i (\zeta_i(x) - \xi_i(x))$. Since $V_i(x)$ is a function of $(\mathcal{D}^i, \boldsymbol{\omega}^i)$ and

$$\mathbb{E}_{\mathcal{D}^i, \boldsymbol{\omega}^i}\left[V_i(x)|\boldsymbol{\omega}^{i-1}\right] = a_t^i \mathbb{E}_{\mathcal{D}^i, \boldsymbol{\omega}^i}\left[\zeta_i(x) - \xi_i(x)|\boldsymbol{\omega}^{i-1}\right] = a_t^i \mathbb{E}_{\mathcal{D}^i, \boldsymbol{\omega}^{i-1}}\left[\mathbb{E}_{\boldsymbol{\omega}^i}\left[\zeta_i(x) - \xi_i(x)|\boldsymbol{\omega}^{i-1}\right]\right] = 0,$$

we have that $\{V_i(x)\}$ is a martingal difference sequence. Further note that

$$|V_i(x)| \leqslant c_i = 2M(\phi + \kappa)|a_t^i|.$$

Then by Azuma's Inequality, for any $\epsilon > 0$,

$$\Pr_{\mathcal{D}^t, \boldsymbol{\omega}^t}\left\{|\sum_{i=1}^t V_i(x)| \geqslant \epsilon\right\} \leqslant 2\exp\left\{-\frac{2\epsilon^2}{\sum_{i=1}^t c_i^2}\right\}$$

which is equivalent as

$$\Pr_{\mathcal{D}^t, \boldsymbol{\omega}^t}\left\{\left(\sum_{i=1}^t V_i(x)\right)^2 \geqslant \ln(2/\delta)\sum_{i=1}^t c_i^2/2\right\} \leqslant \delta.$$

Moreover,

$$\mathbb{E}_{\mathcal{D}^t, \boldsymbol{\omega}^t}\left[\left(\sum_{i=1}^t V_i(x)\right)^2\right] = \int_0^\infty \Pr_{\mathcal{D}^t, \boldsymbol{\omega}^t}\left\{\left(\sum_{i=1}^t V_i(x)\right)^2 \geqslant \epsilon\right\} d\epsilon = \int_0^\infty 2\exp\left\{-\frac{2\epsilon}{\sum_{i=1}^t c_i^2}\right\} d\epsilon = \sum_{i=1}^t c_i^2$$

Since $f_{t+1}(x) - h_{t+1}(x) = \sum_{i=1}^t V_i(x)$, we immediately obtain the two parts of the lemma.  ∎

**Lemma 8**  *Suppose $\gamma_i = \frac{\theta}{i}(1 \leqslant i \leqslant t)$ and $\theta\nu \in (1, 2) \cup \mathbb{Z}_+$. Then we have*

(1) $|a_t^i| \leqslant \frac{\theta}{t}$. *Consequently, $\sum_{i=1}^t (a_t^i)^2 \leqslant \frac{\theta^2}{t}$.*

(2) $\sum_{i=1}^t \gamma_i |a_t^i| \leqslant \begin{cases} \frac{\theta^2(\ln(t)+1)}{t}, & \text{if } \theta\nu \in [1, 2), \\ \frac{\theta^2}{t}, & \text{if } \theta\nu \in [2, +\infty) \cap \mathbb{Z}_+ \end{cases}$.

**Proof**  (1) follows by induction on $i$. $|a_t^t| \leqslant \frac{\theta}{t}$ is trivially true. We have

$$|a_t^i| = |a_t^{i+1} \frac{\gamma_i}{\gamma_{i+1}}(1 - \nu\gamma_{i+1})| = \frac{i+1}{i}|1 - \frac{\nu\theta}{i+1}| \cdot |a_t^{i+1}| = |\frac{i+1-\nu\theta}{i}| \cdot |a_t^{i+1}|.$$

When $\nu\theta \in (1, 2)$, $i - 1 < i + 1 - \nu\theta < i$ for any $i \geqslant 1$, so $|a_t^i| < |a_t^{i+1}| \leqslant \frac{\theta}{t}$. When $\nu\theta \in \mathbb{Z}_+$, if $i > \nu\theta - 1$, then $|a_t^i| < |a_t^{i+1}| \leqslant \frac{\theta}{t}$; if $i \leqslant \nu\theta - 1$, then $|a_t^i| = 0$. For (2), when $\theta\nu \in [1, 2)$,

$$\sum_{i=1}^t \gamma_t |a_t^i| = \sum_{i=1}^t \frac{\theta^2}{i^2} \cdot \frac{i+1-\theta\nu}{i+1} \cdots \frac{t-\theta\nu}{t} \leqslant \sum_{i=1}^t \frac{\theta^2}{i^2} \cdot \frac{i}{i+1} \cdots \frac{t-1}{t} \leqslant \sum_{i=1}^t \frac{\theta^2}{it} \leqslant \frac{\theta^2(\ln(t)+1)}{t}.$$

When $\theta\nu \in \mathbb{Z}_+$ and $2 \leqslant \theta\nu \leqslant t$,

$$\sum_{i=1}^{t} \gamma_t |a_t^i| = \sum_{i=2}^{t} \frac{\theta^2}{i^2} \cdot \frac{i+1-\theta\nu}{i+1} \cdots \frac{t-\theta\nu}{t} \leqslant \sum_{i=1}^{t} \frac{\theta^2}{i^2} \cdot \frac{i-1}{i+1} \cdots \frac{t-2}{t} \leqslant \sum_{i=2}^{t} \frac{\theta^2(i-1)}{it(t-1)} \leqslant \frac{\theta^2}{t}.$$

■

### A.1.2 Error due to random data

**Lemma 9** *Assume $l'(u, y)$ is L-Lipschitz continous in terms of $u \in \mathbb{R}$. Let $f_*$ be the optimal solution to our target problem. Then*

*(i) If we set $\gamma_t = \frac{\theta}{t}$ with $\theta$ such that $\theta\nu \in (1, 2) \cup \mathbb{Z}_+$, then*

$$\mathbb{E}_{\mathcal{D}^t, \boldsymbol{\omega}^t} \left[ \|h_{t+1} - f_*\|_{\mathcal{H}}^2 \right] \leqslant \frac{Q_1^2}{t},$$

*where*

$$Q_1 = \max \left\{ \|f_*\|_{\mathcal{H}}, \frac{Q_0 + \sqrt{Q_0^2 + (2\theta\nu - 1)(1 + \theta\nu)^2 \theta^2 \kappa M^2}}{2\nu\theta - 1} \right\}, Q_0 = 2\sqrt{2}\kappa^{1/2}(\kappa + \phi)LM\theta^2.$$

*Particularly, if $\theta\nu = 1$, we have $Q_1 \leqslant \max \left\{ \|f_*\|_{\mathcal{H}}, 4\sqrt{2}((\kappa + \phi)L + \nu) \cdot \frac{\kappa^{1/2}M}{\nu^2} \right\}$.*

*(ii) If we set $\gamma_t = \frac{\theta}{t}$ with $\theta$ such that $\theta\nu \in \mathbb{Z}_+$ and $t \geqslant \theta\nu$, then with probability at least $1 - 2\delta$ over $(\mathcal{D}^t, \boldsymbol{\omega}^t)$,*

$$\|h_{t+1} - f_*\|_{\mathcal{H}}^2 \leqslant Q_2^2 \frac{\ln(2t/\delta)\ln(t)}{t}.$$

*where*

$$Q_2 = \max \left\{ \|f_*\|_{\mathcal{H}}, Q_0 + \sqrt{Q_0^2 + \kappa M^2 (1 + \theta\nu)^2 (\theta^2 + 16\theta/\nu)} \right\}, Q_0 = 4\sqrt{2}\kappa^{1/2}M\theta(8 + (\kappa + \phi)\theta L).$$

*Particularly, if $\theta\nu = 1$, we have $Q_2 \leqslant \max \left\{ \|f_*\|_{\mathcal{H}}, 8\sqrt{2}((\kappa + \phi)L + 9\nu) \cdot \frac{\kappa^{1/2}M}{\nu^2} \right\}$.*

**Proof** For the sake of simple notations, let us first denote the following three different gradient terms, which are

$$g_t = \xi_t + \nu h_t = l'(f_t(x_t), y_t)k(x_t, \cdot) + \nu h_t,$$
$$\hat{g}_t = \hat{\xi}_t + \nu h_t = l'(h_t(x_t), y_t)k(x_t, \cdot) + \nu h_t,$$
$$\bar{g}_t = \mathbb{E}_{\mathcal{D}_t}[\hat{g}_t] = \mathbb{E}_{\mathcal{D}_t}[l'(h_t(x_t), y_t)k(x_t, \cdot)] + \nu h_t.$$

Note that by our previous definition, we have $h_{t+1} = h_t - \gamma_t g_t, \forall t \geqslant 1$.

Denote $A_t = \|h_t - f_*\|_{\mathcal{H}}^2$. Then we have

$$
\begin{aligned}
A_{t+1} &= \|h_t - f_* - \gamma_t g_t\|_{\mathcal{H}}^2 \\
&= A_t + \gamma_t^2 \|g_t\|_{\mathcal{H}}^2 - 2\gamma_t \langle h_t - f_*, g_t \rangle_{\mathcal{H}} \\
&= A_t + \gamma_t^2 \|g_t\|_{\mathcal{H}}^2 - 2\gamma_t \langle h_t - f_*, \bar{g}_t \rangle_{\mathcal{H}} + 2\gamma_t \langle h_t - f_*, \bar{g}_t - \hat{g}_t \rangle_{\mathcal{H}} + 2\gamma_t \langle h_t - f_*, \hat{g}_t - g_t \rangle_{\mathcal{H}}
\end{aligned}
$$

Because of the strongly convexity of (1) and optimality condition, we have

$$\langle h_t - f_*, \bar{g}_t \rangle_{\mathcal{H}} \geqslant \nu \|h_t - f_*\|_{\mathcal{H}}^2$$

Hence, we have

$$A_{t+1} \leqslant (1 - 2\gamma_t \nu)A_t + \gamma_t^2 \|g_t\|_{\mathcal{H}}^2 + 2\gamma_t \langle h_t - f_*, \bar{g}_t - \hat{g}_t \rangle_{\mathcal{H}} + 2\gamma_t \langle h_t - f_*, \hat{g}_t - g_t \rangle_{\mathcal{H}}, \forall t \geqslant 1 \quad (8)$$

*Proof for (i):* Let us denote $\mathcal{M}_t = \|g_t\|_{\mathcal{H}}^2, \mathcal{N}_t = \langle h_t - f_*, \bar{g}_t - \hat{g}_t \rangle_{\mathcal{H}}, \mathcal{R}_t = \langle h_t - f_*, \hat{g}_t - g_t \rangle_{\mathcal{H}}$. We first show that $\mathcal{M}_t, \mathcal{N}_t, \mathcal{R}_t$ are bounded. Specifically, we have for $t \geqslant 1$,

(1) $\mathcal{M}_t \leqslant \kappa M^2 (1 + \nu c_t)^2$, where $c_t = \sqrt{\sum_{i,j=1}^{t-1} |a_{t-1}^i| \cdot |a_{t-1}^j|}$ for $t \geqslant 2$ and $c_1 = 0$;

(2) $\mathbb{E}_{\mathcal{D}^t,\boldsymbol{\omega}^t}[\mathcal{N}_t] = 0$;

(3) $\mathbb{E}_{\mathcal{D}^t,\boldsymbol{\omega}^t}[\mathcal{R}_t] \leqslant \kappa^{1/2} L B_{1,t} \sqrt{\mathbb{E}_{\mathcal{D}^{t-1},\boldsymbol{\omega}^{t-1}}[A_t]}$, where $B_{1,t}^2 := 4M^2(\kappa+\phi)^2 \sum_{i=1}^{t-1}|a_{t-1}^i|^2$ for $t \geqslant 2$ and $B_{1,1} = 0$;

We prove these results separately in Lemma 10 below. Let us denote $e_t = \mathbb{E}_{\mathcal{D}^{t-1},\boldsymbol{\omega}^{t-1}}[A_t]$, given the above bounds, we arrive at the following recursion,

$$e_{t+1} \leqslant (1 - 2\gamma_t \nu)e_t + \kappa M^2 \gamma_t^2 (1 + \nu c_t)^2 + 2\kappa^{1/2} L \gamma_t B_{1,t} \sqrt{e_t}. \tag{9}$$

When $\gamma_t = \theta/t$ with $\theta$ such that $\theta\nu \in (1,2) \cup \mathbb{Z}_+$, from Lemma 8, we have $|a_t^i| \leqslant \frac{\theta}{t}, \forall 1 \leqslant i \leqslant t$. Consequently, $c_t \leqslant \theta$ and $B_{1,t}^2 \leqslant 4M^2(\kappa+\phi)\frac{\theta^2}{t-1}$. Applying these bounds leads to the refined recursion as follows

$$e_{t+1} \leqslant \left(1 - \frac{2\nu\theta}{t}\right)e_t + \kappa M^2\frac{\theta^2}{t^2}(1+\nu\theta)^2 + 2\kappa^{1/2}L\frac{\theta}{t}\sqrt{4M^2(\kappa+\phi)^2\frac{\theta^2}{t-1}}\sqrt{e_t}$$

that can be further written as

$$e_{t+1} \leqslant \left(1 - \frac{2\nu\theta}{t}\right)e_t + \frac{\beta_1}{t}\sqrt{\frac{e_t}{t}} + \frac{\beta_2}{t^2},$$

where $\beta_1 = 4\sqrt{2}\kappa^{1/2}LM(k+\phi)\theta^2$ and $\beta_2 = \kappa M^2(1+\nu\theta)^2\theta^2$. Invoking Lemma 14 with $\eta = 2\theta\nu > 1$, we obtain

$$e_t \leqslant \frac{Q_1^2}{t},$$

where $Q_1 = \max\left\{\|f_*\|_{\mathcal{H}}, \frac{Q_0+\sqrt{Q_0^2+(2\theta\nu-1)(1+\theta\nu)^2\theta^2\kappa M^2}}{2\nu\theta-1}\right\}$, and $Q_0 = 2\sqrt{2}\kappa^{1/2}(\kappa+\phi)LM\theta^2$.

*Proof for $(ii)$:* Cumulating equations (8) with $i = 1,\ldots t$, we end up with the following inequality

$$\begin{aligned}
A_{t+1} \leqslant {} & \textstyle\prod_{i=1}^t(1-2\gamma_i\nu)A_1 + 2\sum_{i=1}^t\gamma_i\prod_{j=i+1}^t(1-2\nu\gamma_j)\langle h_i - f_*, \bar{g}_i - \hat{g}_i\rangle_{\mathcal{H}} \\
& + 2\sum_{i=1}^t\gamma_i\prod_{j=i+1}^t(1-2\nu\gamma_j)\langle h_i - f_*, \hat{g}_i - g_i\rangle_{\mathcal{H}} + \sum_{i=1}^t\gamma_i^2\prod_{j=i+1}^t(1-2\nu\gamma_j)\|g_i\|_{\mathcal{H}}^2
\end{aligned} \tag{10}$$

Let us denote $b_t^i = \gamma_i\prod_{j=i+1}^t(1-2\nu\gamma_j), 1 \leqslant i \leqslant t$, the above inequality is equivalent as

$$A_{t+1} \leqslant \prod_{i=1}^t(1-2\gamma_i\nu)A_1 + \sum_{i=1}^t\gamma_i b_t^i\mathcal{M}_i + 2\sum_{i=1}^t b_t^i\mathcal{N}_i + 2\sum_{i=1}^t b_t^i\mathcal{R}_i$$

We first show that

(4) for any $0 < \delta < 1/e$ and $t \geqslant 4$, with probability $1 - \delta$ over $(\mathcal{D}^t, \boldsymbol{\omega}^t)$,

$$\textstyle\sum_{i=1}^t b_t^i\mathcal{N}_i \leqslant 2\max\left\{4\kappa^{1/2}M\sqrt{\sum_{i=1}^t(b_t^i)^2A_i},\ \max_i|b_t^i|\cdot C_0\sqrt{\ln(\ln(t)/\delta)}\right\}\sqrt{\ln(\ln(t)/\delta)},$$

where $C_0 = \frac{4\max_{1\leqslant i\leqslant t}\mathcal{M}_i}{\nu}$.

(5) for any $\delta > 0$, with probability $1 - \delta$ over $(\mathcal{D}^t, \boldsymbol{\omega}^t)$,

$$\textstyle\sum_{i=1}^t b_t^i\mathcal{R}_i \leqslant \sum_{i=1}^t b_t^i\kappa^{1/2}L\hat{B}_{2,i}\sqrt{A_i},$$

where $\hat{B}_{2,i}^2 = 2M^2(\kappa+\phi)^2\ln\left(\frac{2t}{\delta}\right)\sum_{j=1}^{i-1}|a_{i-1}^j|^2$.

Again, the proofs of these results are given separately in Lemma 10. Applying the above bounds leads to the refined recursion as follows,

$$\begin{aligned}
A_{t+1} \quad \leqslant \quad & \prod_{i=1}^t(1-2\gamma_i\nu)A_1 + \sum_{i=1}^t\gamma_i b_t^i\mathcal{M}_i + 2\sum_{i=1}^t b_t^i\kappa^{1/2}LB_{2,i}\sqrt{A_i} \\
& + 4\max\left\{4\kappa^{1/2}M\sqrt{\sum_{i=1}^t(b_t^i)^2A_i},\ \max_i|b_t^i|\cdot C_0\sqrt{\ln(\ln(t)/\delta)}\right\}\sqrt{\ln(\ln(t)/\delta)}
\end{aligned}$$

with probability $1 - 2\delta$. When $\gamma_t = \theta/t$ with $\theta$ such that $\theta\nu \in \mathbb{Z}_+$, with similar reasons in Lemma 8, we have $|b_t^i| \leqslant \frac{\theta}{t}, 1 \leqslant i \leqslant t$ and also we have $\prod_{i=1}^{t}(1 - 2\gamma_i\nu) = \prod_{i=1}^{\theta\nu-1}(1 - 2\frac{\theta\nu}{i})\prod_{i=\theta\nu+1}^{t}(1 - 2\frac{\theta\nu}{i})(1 - 2\frac{\theta\nu}{\theta\nu}) = 0$, and $\sum_{i=1}^{t} \gamma_i b_t^i \leqslant \frac{\theta^2}{t}$. Therefore, we can rewrite the above recursion as

$$A_{t+1} \leqslant \frac{\beta_1}{t} + \beta_2\sqrt{\ln(2t/\delta)} \cdot \sum_{i=1}^{t} \frac{\sqrt{A_i}}{t\sqrt{i}} + \beta_3\sqrt{\ln(\ln(t)/\delta)}\frac{\sqrt{\sum_{i=1}^{t} A_i}}{t} + \beta_4\ln(\ln(t/\delta))\frac{1}{t} \qquad (11)$$

where $\beta_1 = \kappa M^2(1 + \nu\theta)^2\theta^2$, $\beta_2 = 2\sqrt{2}\kappa^{1/2}LM(\kappa + \phi)\theta^2$, $\beta_3 = 16\kappa^{1/2}M\theta$, $\beta_4 = 16\kappa M^2(1 + \theta\nu)^2\theta/\nu$. Invoking Lemma 15, we obtain

$$A_{t+1} \leqslant \frac{Q_2^2\ln(2t/\delta)\ln^2(t)}{t},$$

with the specified $Q_2$. ∎

**Lemma 10** *In this lemma, we prove the inequalities (1)–(5) in Lemma 9.*

**Proof** Given the definitions of $\mathcal{M}_t, \mathcal{N}_t, \mathcal{R}_t$ in Lemma 9, we have

(1) $\mathcal{M}_t \leqslant \kappa M^2(1 + \nu\sqrt{\sum_{i,j=1}^{t-1} |a_{t-1}^i| \cdot |a_{t-1}^j|})^2$;
This is because
$$\mathcal{M}_t = \|g_t\|_{\mathcal{H}}^2 = \|\xi_t + \nu h_t\|_{\mathcal{H}}^2 \leqslant (\|\xi_t\|_{\mathcal{H}} + \nu\|h_t\|_{\mathcal{H}})^2.$$
We have
$$\|\xi_t\|_{\mathcal{H}} = \|l'(f_t(x_t), y_t)k(x_t, \cdot)\|_{\mathcal{H}} \leqslant \kappa^{1/2}M,$$
and
$$\|h_t\|_{\mathcal{H}}^2 = \sum_{i=1}^{t-1}\sum_{j=1}^{t-1} a_{t-1}^i a_{t-1}^j l'(f_i(x_i), y_i)l'(f_j(x_j), y_j)k(x_i, x_j)$$
$$\leqslant \kappa M^2 \sum_{i=1}^{t-1}\sum_{j=1}^{t-1} |a_{t-1}^i| \cdot |a_{t-1}^j|.$$

(2) $\mathbb{E}_{\mathcal{D}^t, \boldsymbol{\omega}^t}[\mathcal{N}_t] = 0$;
This is because $\mathcal{N}_t = \langle h_t - f_*, \bar{g}_t - \hat{g}_t\rangle_{\mathcal{H}}$,
$$\begin{aligned}\mathbb{E}_{\mathcal{D}^t, \boldsymbol{\omega}^t}[\mathcal{N}_t] &= \mathbb{E}_{\mathcal{D}^{t-1}, \boldsymbol{\omega}^t}\left[\mathbb{E}_{D_t}\left[\langle h_t - f_*, \bar{g}_t - \hat{g}_t\rangle_{\mathcal{H}}|\mathcal{D}^{t-1}, \boldsymbol{\omega}^t\right]\right] \\ &= \mathbb{E}_{\mathcal{D}^{t-1}, \boldsymbol{\omega}^t}\left[\langle h_t - f_*, \mathbb{E}_{D_t}[\bar{g}_t - \hat{g}_t]\rangle_{\mathcal{H}}\right] \\ &= 0.\end{aligned}$$

(3) $\mathbb{E}_{\mathcal{D}^t, \boldsymbol{\omega}^t}[\mathcal{R}_t] \leqslant \kappa^{1/2}LB_{1,t}\sqrt{\mathbb{E}_{\mathcal{D}^{t-1}, \boldsymbol{\omega}^{t-1}}[A_t]}$, where $B_{1,t}^2 := 4M^2(\kappa + \phi)^2\sum_{i=1}^{t-1} |a_{t-1}^i|^2$;
This is because $\mathcal{R}_t = \langle h_t - f_*, \hat{g}_t - g_t\rangle_{\mathcal{H}}$,
$$\begin{aligned}\mathbb{E}_{\mathcal{D}^t, \boldsymbol{\omega}^t}[\mathcal{R}_t] &= \mathbb{E}_{\mathcal{D}^t, \boldsymbol{\omega}^t}\left[\langle h_t - f_*, \hat{g}_t - g_t\rangle_{\mathcal{H}}\right] \\ &= \mathbb{E}_{\mathcal{D}^t, \boldsymbol{\omega}^t}\left[\langle h_t - f_*, [l'(f_t(x_t), y_t) - l'(h_t(x_t), y_t)]k(x_t, \cdot)\rangle_{\mathcal{H}}\right] \\ &\leqslant \mathbb{E}_{\mathcal{D}^t, \boldsymbol{\omega}^t}\left[|l'(f_t(x_t), y_t) - l'(h_t(x_t), y_t)| \cdot \|k(x_t, \cdot)\|_{\mathcal{H}} \cdot \|h_t - f_*\|_{\mathcal{H}}\right] \\ &\leqslant \kappa^{1/2}L \cdot \mathbb{E}_{\mathcal{D}^t, \boldsymbol{\omega}^t}\left[|f_t(x_t) - h_t(x_t)|\|h_t - f_*\|_{\mathcal{H}}\right] \\ &\leqslant \kappa^{1/2}L\sqrt{\mathbb{E}_{\mathcal{D}^t, \boldsymbol{\omega}^t}|f_t(x_t) - h_t(x_t)|^2}\sqrt{\mathbb{E}_{\mathcal{D}^t, \boldsymbol{\omega}^t}\|h_t - f_*\|_{\mathcal{H}}^2} \\ &\leqslant \kappa^{1/2}LB_{1,t}\sqrt{\mathbb{E}_{\mathcal{D}^{t-1}, \boldsymbol{\omega}^{t-1}}[A_t]}\end{aligned}$$

where the first and third inequalities are due to Cauchy–Schwarz Inequality and the second inequality is due to $L$-Lipschitz continuity of $l'(\cdot, \cdot)$ in the first parameter, and the last step is due to Lemma 7 and the definition of $A_t$.

(4) for any $0 < \delta < 1/e$ and $t \geqslant 4$, with probability at least $1 - \delta$ over $(\mathcal{D}^t, \boldsymbol{\omega}^t)$,

$$\sum_{i=1}^t b_t^i \mathcal{N}_i \leqslant 2 \max \left\{ 4\kappa^{1/2} M \sqrt{\sum_{i=1}^t (b_t^i)^2 A_i}, \ \max_i |b_t^i| \cdot C_0 \sqrt{\ln(\ln(t)/\delta)} \right\} \sqrt{\ln(\ln(t)/\delta)},$$

where $C_0 = \frac{4 \max_{1 \leqslant i \leqslant t} \mathcal{M}_i}{\nu}$.

This result follows directly from Lemma 3 in [29]. Let us define $d_i = d_i(\mathcal{D}^i, \boldsymbol{\omega}^i) := b_t^i \mathcal{N}_i = b_t^i \langle h_i - f_*, \bar{g}_i - \hat{g}_i \rangle_{\mathcal{H}}, 1 \leqslant i \leqslant t$, we have

- $\{d_i\}_{i=1}^t$ is martingale difference sequence since $\mathbb{E}_{\mathcal{D}^i, \boldsymbol{\omega}^i} \left[ \mathcal{N}_i | \mathcal{D}^{i-1}, \boldsymbol{\omega}^{i-1} \right] = 0$.
- $|d_i| \leqslant \max_i |b_t^i| \cdot C_0$, with $C_0 = \frac{4 \max_{1 \leqslant i \leqslant t} \mathcal{M}_i}{\nu}, \forall 1 \leqslant i \leqslant t$.
- $Var(d_i | \mathcal{D}^{i-1}, \boldsymbol{\omega}^{i-1}) \leqslant 4\kappa M^2 |b_t^i|^2 A_i, \forall 1 \leqslant i \leqslant t$.

Plugging in these specific bounds in Lemma 3 in [Alexander et.al., 2012], which is,

$$\Pr \left( \sum_{i=1}^t d_t \geqslant 2 \max\{2\sigma_t, d_{max} \sqrt{\ln(1/\delta)}\} \sqrt{\ln(1/\delta)} \right) \leqslant \ln(t)\delta.$$

where $\sigma_t^2 = \sum_{i=1}^t Var_{i-1}(d_i)$ and $d_{max} = \max_{1 \leqslant i \leqslant t} |d_i|$, we immediately obtain the above inequality as desired.

(5) for any $\delta > 0$, with probability at least $1 - \delta$ over $(\mathcal{D}^t, \boldsymbol{\omega}^t)$,

$$\sum_{i=1}^t b_t^i \mathcal{R}_i \leqslant \sum_{i=1}^t |b_t^i| \kappa^{1/2} L \hat{B}_{2,i} \sqrt{A_i},$$

where $\hat{B}_{2,i}^2 = 2M^2(\kappa + \phi)^2 \ln\left(\frac{2t}{\delta}\right) \sum_{j=1}^{i-1} |a_{i-1}^j|^2$.

This is because, for any $1 \leqslant i \leqslant t$, recall that from analysis in (3), we have $\mathcal{R}_i \leqslant \kappa^{1/2} L |f_t(x_t) - h_t(x_t)| \cdot \|h_t - f_*\|_{\mathcal{H}}$, therefore from Lemma 9,

$$\Pr(b_t^i \mathcal{R}_i \leqslant \kappa^{1/2} L |b_t^i| \hat{B}_{2,i} \sqrt{A_i}) \geqslant \Pr(|f_i(x_i) - h_i(x_i)|^2 \leqslant \hat{B}_{2,i}^2) \geqslant 1 - \delta/t.$$

Taking the sum over $i$, we therefore get

$$\Pr(\sum_{i=1}^t b_t^i \mathcal{R}_i \leqslant \sum_{i=1}^t |b_t^i| \kappa^{1/2} L B_{2,i} \sqrt{A_i}) \geqslant 1 - \delta.$$

∎

Applying these lemmas immediately gives us Theorem 4 and Theorem 5, which implies pointwise distance between the solution $f_{t+1}(\cdot)$ and $f_*(\cdot)$. Now we prove similar bounds in the sense of $L_\infty$ and $L_2$ distance.

## A.2 $L_\infty$ distance, $L_2$ distance, and generalization bound

**Corollary 11 ($L_\infty$ distance)** *Theorem 4 also implies a bound in $L_\infty$ sense, namely,*

$$\mathbb{E}_{\mathcal{D}^t, \boldsymbol{\omega}^t} \|f_{t+1} - f_*\|_\infty^2 \leqslant \frac{2C^2 + 2\kappa Q_1^2}{t}.$$

*Consequently, for the average solution $\hat{f}_{t+1}(\cdot) := \frac{1}{t} \sum_{i=1}^t f_i(\cdot)$, we also have*

$$\mathbb{E}_{\mathcal{D}^t, \boldsymbol{\omega}^t} \|\hat{f}_{t+1} - f_*\|_\infty^2 \leqslant \frac{(2C^2 + 2\kappa Q_1^2)(\ln(t) + 1)}{t}.$$

This is because $\|f_{t+1} - f_*\|_\infty = \max_{x \in \mathcal{X}} |f_{t+1}(x) - f_*(x)| = |f_{t+1}(x_*) - f_*(x_*)|$, where $x_* \in \mathcal{X}$ always exists since $\mathcal{X}$ is closed and bounded. Note that the result for average solution can be improved without log factor using more sophisticated analysis (see also reference in [29]).

**Corollary 12 ($L_2$ distance)** *With the choices of $\gamma_t$ in Lemma 9, we have*

(i) $\mathbb{E}_{\mathcal{D}^t, \boldsymbol{\omega}^t} \|f_{t+1} - f_*\|_2^2 \leqslant \frac{2C^2 + 2\kappa Q_1^2}{t}$,

(ii) $\|f_{t+1} - f_*\|_2^2 \leqslant \frac{C^2 \ln(8\sqrt{e}t/\delta) + 2\kappa Q_2^2 \ln(2t/\delta) \ln^2(t)}{t}$, *with probability at least $1 - 3\delta$ over $(\mathcal{D}^t, \boldsymbol{\omega}^t)$.*

**Proof** *(i)* follows directly from Theorem 4. *(ii)* can be proved as follows. First, we have

$$\|f_{t+1} - f*\|_2^2 = \mathbb{E}_x|f_{t+1}(x) - f_*(x)|^2 \leqslant 2\mathbb{E}_x|f_{t+1}(x) - h_{t+1}(x)|^2 + 2\kappa\|h_{t+1} - f_*\|_{\mathcal{H}}.$$

From Lemma 9, with probability at least $1 - 2\delta$, we have

$$\|h_{t+1} - f_*\|_{\mathcal{H}}^2 \leqslant \frac{Q_2^2 \ln(2t/\delta) \ln^2(t)}{t}. \tag{12}$$

From Lemma 7, for any $x \in \mathcal{X}$, we have

$$\Pr_{\mathcal{D}^t, \boldsymbol{\omega}^t}\left\{ |f_{t+1}(x) - h_{t+1}(x)|^2 \geqslant \frac{2(\kappa+\phi)^2 M^2 \ln(\frac{2}{\epsilon})\theta^2}{t} \right\} \leqslant \epsilon.$$

Since $C^2 = 4(\kappa + \phi)^2 M^2 \theta^2$, the above inequality can be writen as

$$\Pr_{\mathcal{D}^t, \boldsymbol{\omega}^t}\left\{ |f_{t+1}(x) - h_{t+1}(x)|^2 \geqslant \frac{C^2 \ln(\frac{2}{\epsilon})}{2t} \right\} \leqslant \epsilon.$$

which leads to

$$\Pr_{x \sim \mathbb{P}(x)} \Pr_{\mathcal{D}^t, \boldsymbol{\omega}^t}\left\{ |f_{t+1}(x) - h_{t+1}(x)|^2 \geqslant \frac{C^2 \ln(\frac{2}{\epsilon})}{2t} \right\} \leqslant \epsilon.$$

By Fubini's theorem and Markov's inequality, we have

$$\Pr_{\mathcal{D}^t, \boldsymbol{\omega}^t}\left\{ \Pr_{x \sim \mathbb{P}(x)}\left\{ |f_{t+1}(x) - h_{t+1}(x)|^2 \geqslant \frac{C^2 \ln(\frac{2}{\epsilon})}{2t} \right\} \geqslant \frac{\epsilon}{\delta} \right\} \leqslant \delta.$$

From the analysis in Lemma 7, we also have that $|f_{t+1}(x) - h_{t+1}(x)| \leqslant C^2$. Therefore, with probability at least $1 - \delta$ over $(\mathcal{D}^t, \boldsymbol{\omega}^t)$, we have

$$\mathbb{E}_{x \sim \mathbb{P}(x)}[|f_{t+1}(x) - h_{t+1}(x)|^2] \leqslant \frac{C^2 \ln(\frac{2}{\epsilon})}{2t}(1 - \frac{\epsilon}{\delta}) + C^2 \frac{\epsilon}{\delta}$$

Let $\epsilon = \frac{\delta}{4t}$, we have

$$\mathbb{E}_{x \sim \mathbb{P}(x)}[|f_{t+1}(x) - h_{t+1}(x)|^2] \leqslant \frac{C^2}{2t}(\ln(8t/\delta) + \frac{1}{2}) = \frac{C^2 \ln(8\sqrt{e}t/\delta)}{2t}. \tag{13}$$

Summing up equation (13) and (12), we have

$$\|f_{t+1} - f_*\|_2^2 \leqslant \frac{C^2 \ln(8\sqrt{e}t/\delta) + 2\kappa Q_2^2 \ln(2t/\delta) \ln^2(t)}{t}$$

as desired. ∎

From the bound on $L_2$ distance, we can immediately get the generalization bound.

**Theorem 6 (Generalization bound)** *Let the true risk be $R_{true}(f) = \mathbb{E}_{(x,y)}[l(f(x), y)]$. Then with probability at least $1 - 3\delta$ over $(\mathcal{D}^t, \boldsymbol{\omega}^t)$, and $C$ and $Q_2$ defined as previously*

$$R_{true}(f_{t+1}) - R_{true}(f_*) \leqslant \frac{(C\sqrt{\ln(8\sqrt{e}t/\delta)} + \sqrt{2\kappa}Q_2\sqrt{\ln(2t/\delta)}\ln(t))L}{\sqrt{t}}.$$

**Proof** By the Lipschitz continuity of $l(\cdot, y)$ and Jensen's Inequality, we have

$$R_{true}(f_{t+1}) - R_{true}(f_*) \leqslant L\mathbb{E}_x|f_{t+1}(x) - f_*(x)| \leqslant L\sqrt{\mathbb{E}_x|f_{t+1}(x) - f_*(x)|^2} = L\|f_{t+1} - f_*\|_2.$$

Then the theorem follows from Corollary 12. ∎

## A.3  Suboptimality

For comprehensive purposes, we also provide the $O(1/t)$ bound for suboptimality.

**Corollary 13**  *If we set $\gamma_t = \frac{\theta}{t}$ with $\theta\nu = 1$, then the average solution $\hat{f}_{t+1} := \frac{1}{t}\sum_{i=1}^{t} f_i$ satisfies*

$$R(\mathbb{E}_{\mathcal{D}^t,\boldsymbol{\omega}^t}[\hat{f}_{t+1}]) - R(f_*) \leqslant \frac{Q(\ln(t)+1)}{t}.$$

*where $Q = (4\kappa M^2 + 2\sqrt{2}\kappa^{1/2}LM(\kappa+\phi)Q_1)/\nu$, with $Q_1$ defined as in Lemma 9.*

**Proof**  From the anallysis in Lemma 9,we have

$$\langle h_t - f_*, \bar{g}_t\rangle_{\mathcal{H}} = \frac{1}{2\gamma_t}A_t - \frac{1}{2\gamma_t}A_{t+1} + \gamma_t\mathcal{M}_t + \mathcal{N}_t + \mathcal{R}_t$$

Invoking strongly convexity of $R(f)$, we have $\langle h_t - f_*, \bar{g}_t\rangle \geqslant R(h_t) - R(f_*) + \frac{\nu}{2}\|h_t - f_*\|_{\mathcal{H}}^2$. Taking expectaion on both size and use the bounds in last lemma, we have

$$\mathbb{E}_{\mathcal{D}^t,\boldsymbol{\omega}^t}[R(h_t) - R(f_*)] \leqslant (\frac{1}{2\gamma_t} - \frac{\nu}{2})e_t - \frac{1}{2\gamma_t}e_{t+1} + \gamma_t\kappa M^2(1+\nu c_t)^2 + \kappa^{1/2}LB_{1,t}\sqrt{e_t}$$

Assume $\gamma_t = \frac{\theta}{t}$ with $\theta = \frac{1}{\nu}$, then cumulating the above inequalities leads to

$$\sum_{i=1}^{t}\mathbb{E}_{\mathcal{D}^t,\boldsymbol{\omega}^t}[R(h_i) - R(f_*)] \leqslant \sum_{i=1}^{t}\gamma_i\kappa M^2(1+\nu c_i)^2 + \sum_{i=1}^{t}\kappa^{1/2}LB_{1,i}\sqrt{e_i}$$

which can be further bounded by

$$
\begin{aligned}
\sum_{i=1}^{t}\mathbb{E}_{\mathcal{D}^t,\boldsymbol{\omega}^t}[R(h_i) - R(f_*)] &\leqslant \sum_{i=1}^{t}\gamma_i\kappa M^2(1+\nu c_i)^2 + \sum_{i=1}^{t}\kappa^{1/2}LB_{1,i}\sqrt{e_i} \\
&\leqslant \frac{4\kappa M^2}{\nu}\sum_{i=1}^{t}\frac{1}{i} + \frac{2\sqrt{2}\kappa^{1/2}LM(\kappa+\phi)}{\nu}\sum_{i=1}^{t}\sqrt{\frac{e_i}{i}} \\
&\leqslant \frac{4\kappa M^2}{\nu}(\ln(t)+1) + \frac{2\sqrt{2}\kappa^{1/2}LM(\kappa+\phi)}{\nu}Q_1(\ln(t)+1) \\
&= \frac{Q(\ln(t)+1)}{t}
\end{aligned}
$$

By convexity, we have $\mathbb{E}_{\mathcal{D}^t,\boldsymbol{\omega}^t}[R(\hat{h}_{t+1}) - R(f_*)] \leqslant \frac{Q(\ln(t)+1)}{t}$. The corollary then follows from the fact that $\mathbb{E}_{\mathcal{D}^t,\boldsymbol{\omega}^t}[\hat{f}_{t+1}] = \mathbb{E}_{\mathcal{D}^t,\boldsymbol{\omega}^t}[\hat{h}_{t+1}]$ and $R(\mathbb{E}_{\mathcal{D}^t,\boldsymbol{\omega}^t}[\hat{h}_{t+1}]) \leqslant \mathbb{E}_{\mathcal{D}^t,\boldsymbol{\omega}^t}[R(\hat{h}_{t+1})]$. ∎

## A.4  Technical lemma for recursion bounds

**Lemma 14**  *Suppose the sequence $\{\Gamma_t\}_{t=1}^{\infty}$ satisfies $\Gamma_1 \geqslant 0$, and $\forall t \geqslant 1$*

$$\Gamma_{t+1} \leqslant \left(1 - \frac{\eta}{t}\right)\Gamma_t + \frac{\beta_1}{t\sqrt{t}}\sqrt{\Gamma_t} + \frac{\beta_2}{t^2},$$

*where $\eta > 1, \beta_1, \beta_2 > 0$. Then $\forall t \geqslant 1$,*

$$\Gamma_t \leqslant \frac{R}{t}, \text{ where } R = \max\left\{\Gamma_1, R_0^2\right\}, R_0 = \frac{\beta_1 + \sqrt{\beta_1^2 + 4(\eta-1)\beta_2}}{2(\eta-1)}.$$

**Proof**  The proof follows by induction. When $t = 1$, it always holds true by the definition of $R$. Assume the conclusion holds true for $t$ with $t \geqslant 1$, i.e., $\Gamma_t \leqslant \frac{R}{t}$, then we have

$$
\begin{aligned}
\Gamma_{t+1} &\leqslant \left(1 - \frac{\eta}{t}\right)\Gamma_t + \frac{\beta_1}{t\sqrt{t}}\sqrt{\Gamma_t} + \frac{\beta_2}{t^2} \\
&= \frac{R}{t} - \frac{\eta R - \beta_1\sqrt{R} - \beta_2}{t^2} \\
&\leqslant \frac{R}{t+1} + \frac{R}{t(t+1)} - \frac{\eta R - \beta_1\sqrt{R} - \beta_2}{t^2} \\
&\leqslant \frac{R}{t+1} - \frac{1}{t^2}\left[-R + \eta R - \beta_1\sqrt{R} - \beta_2\right] \\
&\leqslant \frac{R}{t+1}
\end{aligned}
$$

where the last step can be verified as follows.

$$(\eta - 1)R - \beta_1\sqrt{R} - \beta_2 = (\eta - 1)\left[\sqrt{R} - \frac{\beta_1}{2(\eta-1)}\right]^2 - \frac{\beta_1^2}{4(\eta-1)} - \beta_2$$

$$\geqslant (\eta - 1)\left[R_0 - \frac{\beta_1}{2(\eta-1)}\right]^2 - \frac{\beta_1^2}{4(\eta-1)} - \beta_2 \geqslant 0$$

where the last step follows from the defintion of $R_0$. ∎

**Lemma 15** *Suppose the sequence $\{\Gamma_t\}_{t=1}^{\infty}$ satisfies*

$$\Gamma_{t+1} \leqslant \frac{\beta_1}{t} + \beta_2\sqrt{\ln(2t/\delta)} \cdot \sum_{i=1}^{t} \frac{\sqrt{\Gamma_i}}{t\sqrt{i}} + \beta_3\sqrt{\ln(\ln(t)/\delta)}\frac{\sqrt{\sum_{i=1}^{t}\Gamma_i}}{t} + \beta_4\ln(\ln(t/\delta))\frac{1}{t}$$

*where $\beta_1, \beta_2, \beta_3, \beta_4 > 0$ and $\delta \in (0, 1/e)$. Then $\forall 1 \leqslant j \leqslant t(t \geqslant 4)$,*

$$\Gamma_j \leqslant \frac{R\ln(2t/\delta)\ln^2(t)}{j}, \text{ where } R = \max\{\Gamma_1, R_0^2\}, R_0 = 2\beta_2 + 2\sqrt{2}\beta_3 + \sqrt{(2\beta_2 + 2\sqrt{2}\beta_3)^2 + \beta_1 + \beta_4}.$$

**Proof** The proof follows by induction. When $j = 1$ it is trivial. Let us assume it holds true for $1 \leqslant j \leqslant t - 1$, therefore,

$$
\begin{aligned}
\Gamma_{j+1} &\leqslant \frac{\beta_1}{j} + \beta_2\sqrt{\ln(2j/\delta)} \cdot \sum_{i=1}^{j} \frac{\sqrt{\Gamma_i}}{j\sqrt{i}} + \beta_3\sqrt{\ln(\ln(j)/\delta)}\frac{\sqrt{\sum_{i=1}^{j}\Gamma_i}}{j} + \beta_4\ln(\ln(j/\delta))\frac{1}{j}\\
&\leqslant \frac{\beta_1}{j} + \beta_2\sqrt{\ln(2j/\delta)}/j \cdot \sum_{i=1}^{j} \frac{\sqrt{R\ln(2t/\delta)\ln^2(t)}}{i}\\
&\quad + \beta_3\sqrt{\ln(\ln(j)/\delta)}\frac{\sqrt{\sum_{i=1}^{j}R\ln(2t/\delta)\ln^2(t)/i}}{j} + \beta_4\ln(\ln(j/\delta))\frac{1}{j}\\
&\leqslant \frac{\beta_1}{j} + \beta_2\sqrt{\ln(2j/\delta)}/j\sqrt{R\ln(2t/\delta)\ln^2(t)(1+\ln(j))}\\
&\quad + \beta_3\sqrt{\ln(\ln(j)/\delta)}/j\sqrt{R\ln(2t/\delta)\ln^2(t)}\sqrt{\ln(j)+1} + \beta_4\ln(\ln(j/\delta))\frac{1}{j}\\
&\leqslant \frac{\beta_1}{j} + 2\beta_2\sqrt{R}\ln(2t/\delta)\ln^2(t)/j + \sqrt{2}\beta_3\sqrt{R}\ln(2t/\delta)\ln^2(t)/j + \beta_4\ln(2t/\delta)\frac{1}{j}\\
&\leqslant (2\beta_2 + \sqrt{2}\beta_3)\sqrt{R}\frac{\ln(2t/\delta)\ln^2(t)}{j} + (\beta_1 + \beta_4\ln(2t/\delta))\frac{1}{j}\\
&\leqslant \frac{\ln(2t/\delta)\ln^2(t)}{j}[(2\beta_2 + \sqrt{2}\beta_3)\sqrt{R} + \frac{\beta_1}{2} + \frac{\beta_4}{2})
\end{aligned}
$$

Since $\sqrt{R} \geqslant 2\beta_2 + 2\sqrt{2}\beta_3 + \sqrt{(2\beta_2 + 2\sqrt{2}\beta_3)^2 + \beta_1 + \beta_4}$, we have $(2\beta_2 + 2\sqrt{2}\beta_3)\sqrt{R} + \frac{\beta_1}{2} + \frac{\beta_4}{2} \leqslant R/2$.
Hence, $\Gamma_{j+1} \leqslant \frac{R\ln(2t/\delta)\ln^2(t)}{j+1}$. ∎

# B   Experiment Details

We have illustrated the comparison on (2) – (4) with the alternative algorithms stopping when they pass through the entire dataset once in main text. To verify the theoretical guarantee empirically, we conduct experiment on a 2D regression synthetic dataset (1) with $2^{20}$ data points using SC1 stopping criterion. To further demonstrate the advantages of the proposed algorithm in computational cost, we also conduct experiments on datasets (2) – (4) running the competitors within the same time budget as the proposed algorithm (SC2).

Table 1: Datasets

|     | Name | Model | # of samples | Input dim | Output range | Virtual |
|-----|------|-------|--------------|-----------|--------------|---------|
| (1) | Synthetic | K-ridge | $2^{20}$ | 2 | $[-1, 1.3]$ | no |
| (2) | Adult | K-SVM | 32K | 123 | $\{-1, 1\}$ | no |
| (3) | MNIST 8M 8 vs. 6 [25] | K-SVM | 1.6M | 784 | $\{-1, 1\}$ | yes |
| (4) | Forest | K-SVM | 0.5M | 54 | $\{-1, 1\}$ | no |
| (5) | MNIST 8M [25] | K-logistic | 8M | 1568 | $\{0, \ldots, 9\}$ | yes |
| (6) | CIFAR 10 [26] | K-logistic | 60K | 2304 | $\{0, \ldots, 9\}$ | yes |
| (7) | ImageNet [27] | K-logistic | 1.3M | 9216 | $\{0, \ldots, 999\}$ | yes |
| (8) | QuantumMachine [28] | K-ridge | 6K | 276 | $[-800, -2000]$ | yes |
| (9) | MolecularSpace [28] | K-ridge | 2.3M | 2850 | $[0, 13]$ | no |

## B.1 Detailed experiment setups

**Synthet** In this experiment, we compared the seven algorithms listed in the table for solving the kernel ridge regression problem. We use Gaussian RBF kernel with kernel bandwidth $\sigma$ chosen to be 0.1 times the median of pairwise distances between data points (median trick). The regularization parameter $\nu$ is set to be $10^{-6}$. The batch size and feature block are set to be $2^{10}$.

**Adults** In this experiment, we compared the seven algorithms listed in the table for solving the kernel support vector machine problem. We use Gaussian RBF kernel with kernel bandwidth obtained by median trick. The regularization parameter $\nu$ is set to be $1/(100n)$ where $n$ is the number of training samples. We set the batch size to be $2^6$ and feature block to be $2^5$.

**MNIST 8M 8 vs. 6.** We first reduce the dimension to 50 by PCA and use Gaussian RBF kernel with kernel bandwidth $\sigma = 9.03$ obtained by median trick. The regularization parameter $\nu$ is set to be $1/n$ where $n$ is the number of training samples. We set the batch size to be $2^{10}$ and feature block to be $2^8$.

**Forest.** We use Gaussian RBF kernel with kernel bandwidth obtained by median trick. The regularization parameter $\nu$ is set to be $1/n$ where $n$ is the number of training samples. We set the batch size to be $2^{10}$ and feature block to be $2^8$.

**MNIST 8M.** In this experiment, we compared to a variant of LeNet-5 [30], where all tanh units are replaced with rectified linear units. We also use more convolution filters and a larger fully connected layer. Specifically, the first two convolutions layers have 16 and 32 filters, respectively, and the fully connected layer contains 128 neurons. We used kernel logistic regression for the task. We extracted features from the last max-pooling layer with dimension 1568, and used Gaussian RBF kernel with kernel bandwidth $\sigma$ equaling to four times the median pairwise distance. The regularization parameter $\nu$ is set to be 0.0005.

**CIFAR 10** In this experiment, we compared to a neural net with two convolution layers (after contrast normalization and max-pooling layers) and two local layers that achieves 11% test error[2]. The feature is extracted from the top max-pooling layer from a trained neural net and is of dimension 2304. We used kernel logistic regression for this problem. The kernel bandwidth $\sigma$ for Gaussian RBF kernel is again four times the median pairwise distance. The regularization parameter $\nu$ is set to be 0.0005. We also performed a PCA (without centering) to reduce the dimension to 256 before feeding to our method.

**ImageNet** In this experiment, we compared our algorithm with the neural nets on the ImageNet 2012 dataset, which contains 1.3 million color images from 1000 classes. Each image is of size $256 \times 256$, and we randomly crop a $240 \times 240$ region with random horizontal flipping. The jointly-trained neural net is Alex-net [27]. The feature for our classifier and fixed neural net is from the last pooling layer of the jointly-trained neural net, which is 9216 dimensional. The kernel bandwidth $\sigma$ for Gaussian RBF kernel is again four times the median pairwise distance. The regularization parameter $\nu$ is set to be 0.0005.

**QuantumMachine** In this experiment, we used kernel ridge regression for this problem and compared the performance with the neural network as presented in [28]. First of all a set of randomly sorted coulomb matrices were generated for each molecule and then each dimension of the Coulomb matrix is broken apart into steps and converted to the binary predicates as in [28]. For this experiment 40 set of randomly permuted matrices were generated for each training example and 20 for each test example. Predictions were made by taking average of all prediction made on various coulomb matrices of same molecule. Gaussian kernel with kernel bandwidth

(1) 2D Synthetic Dataset     (2) Convergence Rate     (3) Comparison Accuracy vs. Time

Figure 2: Experimental results for kernel ridge regression on synthetic dataset.

SC2: (1) Adult     (2)MNIST 8M 8 vs. 6     (3) Forest.

Figure 3: Comparison with other kernel SVM solvers on datasets (2) – (4) with stopping criteria SC2.

60 (obtained from median trick) is used for the experiment with $2^{20}$ dimensional random features. In every iteration 2048 dimension of the weight vector is updated with the batch size of 50,000.

**MolecularSpace** In this experiment, we again tried to predict the power conversion efficiency of the molecule using kernel ridge regression. This dataset of 2.3 million molecular motifs was obtained from the Clean Energy Project Database.We used the same feature representation as for "QuantumMachine" dataset [28]. Step size used is 10 and we didn't generate multiple randomly sorted coulomb matrices this time. Kernel bandwidth used for Gaussian RBF kernel used is 290. The dimension of the random features is $2^{20}$. In every iteration 2048 dimensions of the weight vector was updated with batch of 25000.

## B.2 Extra experiments

### B.2.1 Regression Comparisons on Synthetic Dataset

In this section, we compare our approach with alternative algorithms for kernel ridge regression on 2D synthetic dataset and stop the algorithms when they pass through the whole dataset once. The data are generated by

$$ y = \cos(0.5\pi\|x\|_2)\exp(-0.1\pi\|x\|_2) + 0.1e $$

where $x \in [-10, 10]^2$ and $e \sim \mathcal{N}(0, 1)$. The results are shown in Figure 2. In Figure 2(1), we plot the optimal functions generating the data. We justify our proof of the convergence rate in Figure 2(2). The blue dotted line is a convergence rate of $1/t$ as a guide. $\hat{f}_t$ denotes the average solution after $t$-iteration, *i.e.*, $\hat{f}_t(x) = \frac{1}{t}\sum_{i=1}^{t} f_i(x)$. It could be seen that our algorithm indeed converges in the rate of $O(1/t)$. In Figure 2 (3), we compare the algorithms discussed in the Sec. 6 for solving the kernel ridge regression.

The comparison on synthetic dataset demonstrates the advantages of our algorithm clearly. Our algorithm achieves comparable performance with NORMA, which uses full kernel, in similar time but only costs $O(n)$ memory while NORMA costs $O(dn)$. The pegasos and SDCA using $2^8$ random or Nyström features perform worse.

## B.3 Classification Comparisons with Kernel SVM Algorithms with SC2

We evaluate our algorithm solving kernel SVM on three datasets (2)–(4) comparing with other several algorithms listed in Sec. 6 using stopping criteria SC2, *i.e.*, running the competitors within the same time budget as the proposed algorithm and the same experiments settings.

**Adult.** The performances are illustrated in Figure 3(1). Under the same time budget, all the algorithms perform similarly, achieving test error 15%. The reason of flat region of r-pegasos, NORMA and the proposed method on this dataset is that Adult dataset is unbalanced. There are about 24% positive samples while 76% negative samples.

**MNIST 8M 8 vs. 6.** The results are shown in Figure 3(2). Our algorithm achieves the best test error 0.26% using similar training time.

**Forest.** In Figure 3(3), we shows the performances of all algorithms using SC2. NORMA achieve the best error rate, which is about 10%, while our algorithm achieves around 15%, but still much better than all the other alternatives.

As seen from the performance of pegasos and SDCA on Adult and MNIST, using fewer features does not deteriorate the classification error. This might be because there are cluster structures in these two binary classification datasets. Thus, they prefer low rank approximation rather than full kernel. Different from these two datasets, in the forest dataset, algorithms with full kernel, *i.e.*, NORMA and k-SDCA, achieve best performance. With more random features, our algorithm performs much better than pegasos and SDCA under both SC1 and SC2. Our algorithm is preferable for this scenario, *i.e.*, huge dataset with sophisticated decision boundary. Although utilizing full kernel could achieve better performance, the computation and memory requirement for the kernel on huge dataset are costly. To learn the sophisticated boundary while still considering the computational and memory cost, we need to efficiently approximate the kernel in $O(\frac{1}{\epsilon})$ with $O(n)$ random features at least. Our algorithm could handle so many random features efficiently in both computation and memory cost, while for pegasos and SDCA such operation is prohibitive.

## Footnotes

[2]The specification is at https://code.google.com/p/cuda-convnet/