[Reviews · NeurIPS 2014]

Submitted by Assigned_Reviewer_2

The paper proposes to use random functions as an approximation to kernel functions and then proposes to do
stochastic gradient descent. Convergence rates and generalisation bounds are derived. Experimental results on
large datasets are presented.

The idea of introducing random functions to approximate kernel functions and then using SGD is very interesting.
The resultant analysis is not standard and requires careful analysis.
Experimental results are impressive.

The choice of \phi, seems to be closely tied to the Kernel function and maybe difficult to compute for arbitrary kernels.
Can the claims be made more precise to reflect this issue e.g. Translation invariant Kernels can be used instead of kernels

SGD is prone to have high variance leading to more iterations. With one more source of noise the variance can get larger compounding the problem. It would have been nice if the paper would have studied the problem of higher variance.
Summary: This is an interesting paper which could be of interest to Kernel learning folks at NIPS.

Submitted by Assigned_Reviewer_41

The paper proposes a large-scale kernel machine variant based on a combination of stochastic functional gradient descent and kernel random features. It is shown that the resulting estimate of the gradient is unbiased, thereby opening the door to convergence proofs. Generalization bounds are given, and the paper concludes with collection of experiments mainly involving datasets enlarged with virtual examples.

The clarity, organization, and flow of the paper is good (although there are some grammatical errors that should be corrected). It is clear the authors gave some thought to their presentation.
A noteworthy aspect of this submission is that it is thorough, particularly in light of the space constraints. The authors explain where the problem is, how their proposed solution addresses those problems, provide some theory to ensure the suggested algorithm is reasonable, and test their ideas on multiple, relatively realistic datasets against some existing alternatives. The theoretical component may, perhaps, be seen as among the more interesting contributions of the paper (more on why below). The observation that one need only store a random seed and a pseudo-RNG routine to reproduce a sequence of random projections is a nifty one. The experiments constitute a decent illustration of the method, and the authors should be commended for providing the additional experimental detail given in the supplementary material.

A concern with the paper is a possible lack of novelty and significance. Fundamentally, the main idea is to combine two pre-existing ideas: functional gradient descent and random features approximating a (shift-invariant) kernel. Large n is the target problem addressed by both, but the key motivation for using kernel random features, and for increasing complexity, is to allow to reduce test-time storage by exchanging memory for additional computation. However, the paper should address the following important questions around this reasoning:

- It is true that testing requires minimal storage, under the assumption that you only really need to store the RNG seed from which one can then re-generate the random projections assumed during training. But this is also true for the vanilla random features approach anyways.

- The proposed method exchanges low test-time memory footprint for a potentially huge, redundant computational burden associated with repeated random number generation and projection steps. The regime where this becomes acceptable could be for very large n indeed, particularly since random projections are computed anew during training iterations. It could be argued, even without random projections of any kind, that for all but the largest training sets, the test time storage burden is less of a concern. O(n) -- a single pass over the data -- may not be all that bad for many large applications. If random features are used, then a similar argument might be made for storing the projections.

- The functional gradient descent approach comes with its own drawbacks. It may only be useful for more exotic loss functions, since if we use random features alone in the setting where (n is much larger than p), the square loss or hinge loss will result in a very small linear problem with random features alone. SGD may not be necessary.

In light of these questions, the method may not have the strongest practical implications. The theory, if correct, could however serve as an interesting illustration of martingale methods, since the (mathematical) problem is clear, and the goals are relatively self-contained.
Summary: A nicely executed paper. However, questions remain about the incremental nature of the work (seen as a combination of functional SGD plus random features), and about the utility of the idea (many of the advantages appear to apply to both SGD and/or random features separately).

Submitted by Assigned_Reviewer_42

Summary:
The paper introduces an SGD scheme, where not only the error function is approximated by a single sample, but also the kernel itself is approximated using a single sample. While the obtained classifier itself is not a member of the RKHS, the authors proof that the algorithm still converges to the true optimum and give bounds on the generalization error and the convergence rate.
The interesting twist of the algorithm is, that it is not needed to store the samples from the kernels but only the seeds used to generate them. Thus the storage requirements of the method are independent from the dimensionality of the data, at the cost that the samples have to be drawn from the distribution.

The paper is overall well written and the proposed algorithm indeed is a novel and interesting idea. My only concern is in the experimental section. (i have not checked the proofs, but the claims are reasonable enough)

1. without the supplement it is impossible to see what the actual problem sizes used in the experiments are, as only the description of the initial datasets is given.

2. Experiments a) and b) suffer from a very early stopping criterion. Given the running times it is unreasonable to stop after one pass through the dataset as this only measures the efficiency of single iterations. Thus an algorithm which is optimized for fast single iterations will be stopped very early, even though it could still make a lot of progress. A more reasonable stopping criterion would be to take the time used by the newly proposed algorithm (or NORMA) and then run all algorithms for this time budget.
Moreover, given that the experiments appear to be running so fast, I would expect proper median/quantiles or mean/variances over a big set of runs (like a 100 or so)
Summary: Overall a good paper, but the experimental section could be written more clearly and extended.
Author Feedback
Author rebuttal: We appreciate the reviewers for the comments. We are very grateful to the overall positive assessment of our work. We address the reviewers' concerns below:

Reviewer 2:

1. Beyond shift-invariant kernel:
The duality between kernel functions and random processes are generic, and *many examples have been worked out in the literature beyond shift-invariant kernels*. For instance,
-general dot product kernel k(< x,y >), e.g., polynomial kernels [1, 2],
-additive/multiplicative class of homogeneous kernels [3], e.g., Hellinger's, chi^2 and Jensen-Shannon's kernel,
-kernels on Abelian semigroups [4].
These explicit random features can be obtained efficiently via simple sampling processes (see table 1. in [4] for examples). As long as the random features of these kernels satisfy assumption D in our paper, which is indeed true for many kernels listed above, our algorithm and analysis can be applied.
[1]P. Kar&H. Karnick. Random feature maps for dot product kernels. AISTATS12.
[2]N. Pham&R. Pagh. Fast and scalable polynomial kernels via explicit feature maps. KDD13.
[3]A. Vedaldi&A. Zisserman. Efficient additive kernels via explicit feature maps. PAMI12.
[4]J. Yang, et al. Random laplace feature maps for semigroup kernels on histograms. CVPR14.

2. Variance of random features:
We have analyzed the effect of additional variance introduced by random features (see the "Remarks" paragraph, and more details in Lemma 7 and Lemma 10.(1) of the appendix). Surprisingly, *the variance of the random features only contributes additively* to the constant in the final convergence rate. We still get an overall O(1/t) rate of convergence. Moreover, techniques for reducing variance of SGD are available (e.g. [5] and references therein), which could be adapted to improve the constant in the convergence rate.
[5]R. Johnson&T. Zhang. Accelerating stochastic gradient descent using predictive variance reduction. NIPS13.

Reviewer 41:

1. Novelty and significance:
-New streaming nonparametric algorithm with simple update. Our algorithm is designed for really large scale streaming nonparametric problems where the main memory can not hold all data points, and SGD is the method of choice (e.g. Imagenet with n > 10^7 virtual examples and d > 10^3 feature dimensions will require Terabytes of memory). In this regime, the number of parameters of a hypothesis needs to grow linearly with the number data points (n) to achieve full potential of nonparametric methods. Furthermore, solving linear system even for square/hinge loss is not practical (dense matrix, O(n^3) operations). Our algorithm is simple, and *does not need to commit to a preset number of parameters, and it allows the flexibility of the function class to grow as we see more incoming data*.

-Novel analysis of convergence. In hindsight, it seems natural to combine SGD and random features. However, it is *not a-priori clear* whether such combination will result in a convergent algorithm and the additional variance of the random features will drag down the overall convergence rate. We provide a novel and nontrivial analysis involving Hilbert space martingale and a newly proved recurrence relation, and show that the estimator converges to the optimal RKHS function, and the variance of the random features only contributes additively to the constant in the final convergence rate. Both results are the first of the kind in kernel method literature, which could be of independent interest.

-Practical implications. Our algorithm scales kernel methods up to the regimes which are previously dominated by neural nets, and achieves comparable results to neural nets (e.g. Imagenet with n >10^7 virtual examples). Our results suggest that kernel methods, theoretically well-grounded methods, can potentially replace neural nets in many large scale real-world problems where nonparametric estimation are needed.

2. Advantages over vanilla SGD and/or random features:

In our method, the learned function can be updated easily as more incoming data points need to be fitted, and the updated function will enjoy improved guarantee due to the new data. In comparison:

-Vanilla stochastic functional gradient (SGD, ie, NORMA in paper): Need to store all training examples, which incurs a prohibitively high memory of O(nd). Imagenet dataset, n>10^7 virtual examples and d>10^3, immediately imposes a 10^10 memory requirement.

-Vanilla random features (RF): Need to commit to a preset number of random features to start with, and *not directly applicable to streaming setting*. If one wants to introduce more random features as more data arrive, it is not clear how to efficiently update the new model incrementally to obtain the same convergence rate.

-SGD+RF: The same drawback as RF.
-Random block coordinate descent (BC) + RF. The same drawback as RF.
-BC+SGD+RF. The same drawback as RF.

Furthermore, our algorithm (doubly SGD) is the cheapest in terms of per training iteration computation and memory requirement. For instance, to achieve epsilon error (denoted as e), then at iteration t < n (denoted as @t):
Algorithm | Computation@t | Memory@t | Iteration #
Doubly SGD | \Theta(dt + t + t) | \Theta(t) | O(1/e) ~ O(n)
SGD | \Theta(dn + n + n) | \Theta(n) | O(1/e) ~ O(n)
BC+RF | \Theta(dn^2+n^2+n/b)| \Theta(n) | O(log(1/e)) ~ O(log(n))
BC+SGD+RF | \Theta(dn + n + n/b)| \Theta(n) | O(b/e) ~ O(bn)

b: feature block size in BCD
\Theta(.) notation means the same order as (.)

Reviewer 42:

1. We will move more details on experiments to the main text.

2. Stopping criterion:
We did some experiments using the suggested stopping criterion. The conclusion does not change: the doubly SGD achieves comparable or better performance to NORMA and other competitors in the same time budget. We will present them in our final version.